

# Long-range Transport Impacts on Surface Aerosol Concentrations and the Contributions to Haze Events in China: an HTAP2 Multi-Model Study

Xinyi Dong[1], Joshua S. Fu[1], Qingzhao Zhu[1], Jian Sun[1], Jiani Tan[1], Terry Keating[2], Takashi Sekiya[3], Kengo Sudo[3], Louisa Emmons[4], Simone Tilmes[4], Jan Eiof Jonson[5], Michael Schulz[5], Huisheng Bian[6], Mian Chin[7], Yanko Davila[8], Daven Henze[8], Toshihiko Takemura[9], Anna Maria Katarina Benedictow[5], Kan Huang[1, 10]

[1]Department of Civil and Environmental Engineering, The University of Tennessee, Knoxville, Tennessee, USA
[2]Environmental Protection Agency, Applied Science and Education Division, National Center for Environmental Research, Office of Research and Development, Headquarters, Federal Triangle, Washington, DC 20460, USA
[3]Nagoya University, Furo-cho, Chikusa-ku, Nagoya, Japan
[4]Atmospheric Chemistry Observations and Modeling Laboratory, National Center for Atmospheric Research, Boulder, Colorado, USA
[5]Norwegian Meteorological Institute, Oslo, Norway
[6]Goddard Earth Sciences and Technology Center, University of Maryland, Baltimore, MD, USA
[7]Earth Sciences Division, NASA Goddard Space Flight Center, Greenbelt, MD, USA
[8]Department of Mechanical Engineering, University of Colorado, Boulder, CO, USA
[9]Research Institute for Applied Mechanics, Kyushu University, Fukuoka, Japan
[10]Center for Atmospheric Chemistry Study, Department of Environmental Science and Engineering, Fudan University, Shanghai 200433, China

Correspondence to: Joshua S. Fu (jsfu@utk.edu)

## Abstract

Haze has been severely affecting the densely populated areas in China during recent years. While many of the pilot studies have been devoted to investigate the contributions from local anthropogenic emission, limited attention has been paid to the influence from long-range transport. In this study, we use simulations from 6 participating models supplied through the Task Force on Hemispheric Transport of Air Pollution Phase 2 (HTAP2) exercise to investigate the long-range transport impact of Europe and Russia/Belarussia/Ukraine on the surface air quality in East Asia, with special focus on their contributions during the haze episodes over China. The impact of 20% anthropogenic emission perturbation from the source region is extrapolated by a factor of 5 to estimate the full impact. We find that the full impacts from EUR and RBU are $0.99\mu g/m^3$ (3.1%) and $1.32\mu g/m^3$ (4.1%) respectively during haze episodes, while the annual averaged full impacts are only $0.35\mu g/m^3$ (1.7%) and $0.53\mu g/m^3$ (2.6%) respectively. By estimating the aerosol response within and above the planetary boundary layer (PBL), we find that long-range transport within the PBL contributes to 22-38% of the total column density of aerosol response. Comparison with the HTAP Phase 1 (HTAP1) assessment reveals that from 2000 to 2010, the long-range transport from Europe to East Asia has decreased significantly by a factor of 2-10 for surface aerosol mass concentration due to the simultaneous emission reduction in source region and emission increase in the receptor region. By investigating the visibility response, we find that the long-range transport from the Europe and RBU region increases the number of haze events in China by 0.15% and 0.11% respectively, and the North China Plain and southeast China receives 1-3 extra haze days. This study is the first investigation into the contribution of long-range transport to haze in China with multiple model experiments.





## 1. Introduction

Frequent low visibility due to heavy haze has been one of the most important environmental concerns in China during recent years. Long-term monitoring data suggest that visibility degradation has been identified during the past 30 years over North China Plain, Pearl River Delta, and Yangtze River Delta (Fu et al., 2014;Wang et al., 2014a), where more than 40% of the national population is hosted. As the most apparent symptom of air pollution, visibility degradations induced by haze not only interrupt highway and airline operations, but also indicate critical deterioration of public health. The China Ministry of Environmental Protection (MEP) reported that air quality in 265 of the 338 major cities failed to attain the national air quality standard in 2015 (Wang, 2017), and World Bank also suggests that 350,000-400,000 annually premature deaths are attributable to air pollution exposure (WorldBank, 2007) in China.

China haze is usually associated with high concentrations and rapid hygroscopic growth of fine particulate matters (PM). Although only recently has public attention been centered on $PM_{2.5}$ with national level regular measurement data released since 2013, many pilot studies have already been conducted for years to explore the understanding of haze in China. The topics attracted most of the research efforts are focused on: ambient air quality conditions under haze condition (Huang et al., 2012;Wang et al., 2015), spatial distribution and long-term trend of haze in China (Fu et al., 2014), meteorology conditions that favor the formation of haze (Wang et al., 2014a), chemical components and size distributions of aerosols (Guo et al., 2014;Ho et al., 2016;Shen et al., 2017;Yin et al., 2012;Zhang et al., 2012), source apportionment of fine particles during haze episodes (Hua et al., 2015;Wang et al., 2014b;Wang et al., 2014c), and also public health impact of haze (Gao et al., 2017;Tie et al., 2009;Xu et al., 2013).

Although these pilot studies shed light on the understanding of fundamental characteristics of haze, very limited attention has been paid to estimate the contribution from outside the country through long-range transport. Research community has realized the hemispheric transport could also exacerbate regional and even inter-continental air quality problems since early of the 20[th] century, and several international collaborated programs have been initiated to investigate the long-range transport of air pollutants since then. The Task Force on Hemispheric Transport of Air Pollution (TF HTAP) is designated to advance the understanding of inter-continental transport of air pollutants in the Northern Hemisphere (Dentener et al., 2010), Air Quality Modeling Evaluation International Initiative (AQMEII) coordinated by the Joint Research Centre/IES, Environment Canada, and United States Environmental Protection Agency (U.S. EPA) aims at promoting research on transatlantic transport of air pollutants (Rao et al., 2011), the Model Inter-Comparison Study Asia (MICS-Asia) organized by multiple research institutions from East Asia and Southeast Asia countries devoted their efforts to develop the local anthropogenic emissions and estimate the source apportionment of acid depositions and air pollutants transport within Asian (Carmichael et al., 2008). These cooperative efforts pave the way for the research community to probe into the air pollution problem with an international perspective, and enable the exchange of information between countries/regions for keeping in focus of policy needs as well.

The abovementioned prior efforts however, have limited assessment of long-range transport impact on haze which is likely due to the fact that haze is a relatively recent topic and mostly causing problems in China and India. In order to achieve a better air quality condition and reduce the frequency of haze events, China is spending billions to reduce the local anthropogenic emissions (Li and Zhu, 2014;Liu et al., 2015) and trying to stop scarifying the environment benefit for economic growth. However, the background concentrations of PM and the contributions from outside China import of air pollutants to the haze problem,



is poorly documented. Understanding of the long-range transport impact is essential to estimate the background concentrations of air pollutants and estimate the efficiency and effectiveness of local emission control, it is also an important scientific support for policy makers to better organize the international collaborations.

5   In this study, we evaluate the long-range transport impact on haze in China by estimating the PM concentration response and visibility change based on multi-model data provided by the second phase of HTAP (HTAP2). We focused on transport from two source regions designed by the HTAP2 framework: Europe (Tie et al.) and Russia/Belarussia/Ukraine (RBU) since they are the most important upper wind areas with respect to East Asia (EAS) as the receptor region. Modeling framework and baseline evaluation

is described in section2. Results and discussions are summarized in section 3, including the demonstration of long-range transport seasonality, comparison of PM transport above and within the planetary boundary layer (PBL), the assessment of full impact and relative importance of long-range transport, and also the contributions during haze episodes in China. Conclusions are summarized in section 4.

## 2. Method

**2.1 Models, emissions, and simulations configuration**

  The HTAP2 participating models all utilize the same anthropogenic emission inventories for $SO_2$, $NO_x$, CO, non-methane VOC (NMVOC), $NH_3$, $PM_{10}$, $PM_{2.5}$, black carbon (BC) and organic carbon (OC), which are compiled from several regional inventories for the year 2010 with monthly temporal resolution and $0.1°\times0.1°$ grid resolution, with more details reported in Janssens-Maenhout et al.(Janssens-Maenhout

et al., 2015). Emissions of year 2008 and 2009 are also prepared in the same format as that of 2010 through the HTAP2 effort, yet model simulations for these two years are of lower priorities. So in this study we will mainly focus on the 2010 model experiments, but also probe into the inter-annual variability by utilizing the 2008 and 2009 model experiments, which is only available from a few participating models. Emissions from biomass burning and natural sources are not prescribed by the HTAP2 framework, but most of the

participating models used the recommended Global Fire Emission Database version 3 (GFED3) and Model of Emissions of Gases and Aerosols from Nature (MEGAN) for biomass burning and biogenic emissions respectively. To quantify the contribution from each source region with sensitivity simulations, emission perturbation is conducted with all anthropogenic emissions cut off by 20% over the source region. To examine the relative importance of long-range transport as compared to local emission change, emission

perturbation is also performed for the receptor region only. So this study utilizes the simulations from four scenarios: (Guido R. van der Werf) BASE scenario with all baseline emissions; (2) EURALL scenario with all anthropogenic emissions from EUR reduced by 20%, (3) RBUALL scenario with all anthropogenic emissions from RBU reduced by 20%, and (4) EASALL scenario with all anthropogenic emissions from EAS reduced by 20%. Domain configurations of these regions are shown in Fig.1. Note that all model

experiments are conducted at global scale but the analysis of this study will focus on EUR, RBU, and EAS only.

  This study takes input from 6 global models with their grid resolution, meteorology, and references listed in Table 1. These models are selected because they have the simulations from the BASE, EURALL,

RBUALL, and EASALL scenarios of the model level PM mass concentrations. These datasets are essential to estimate surface PM response compare the aerosol transport in different atmosphere layers. Long-range



transport of air pollutants may occur near the planetary boundary layer (PBL) or occur in the upper free troposphere and then descend into the PBL, thus it is necessary to understand the contributions through PBL and free troposphere respectively. Since vertical distribution of suspended particles, especially as relative to cloud height, plays an important role in determining radiative forcing disturbance, investigating

contributions of long-range transport in PBL and free troposphere will also help to examine the long-range transport impact on regional climate, which has been reported in Stjern et al. (Stjern et al., 2016). These participating models have grid resolution around 1×1° and are generally sufficient to demonstrate the broad impacts from one continent to another (Dentener et al., 2010). Although these simulations are relatively coarse as compared to regional modeling, they are able to probe into the transport of upwind pollutants in

both low and high altitudes (Fiore et al., 2009).

## 2.2 Model evaluation

Analyzing the source-receptor (S/R) relationship almost purely relies on model simulations, thus the performances of the models, especially over the source and receptor regions, determine the reliability

of long-range transport assessment. To understand the accuracy and uncertainty of the simulations, measurements from multiple observation networks are employed in this study to evaluate the models performances at EUR, RBU, and EAS regions respectively. Surface observations are collected from four programs: EBAS from the Norwegian Institute for Air Research (NILU, ebas.nilu.no), Air Pollution Index (API) from the China Ministry of Environmental Protection, Acid Deposition Monitoring Network in East

Asia (EANET), and the AERONET (http://aeronet.gsfc.nasa.gov) from NASA. EBAS (Torseth et al., 2012) sites are all located in Europe so the data is used for model evaluation in EUR. API includes $PM_{10}$ concentrations from 86 cities over China (Dong et al., 2016), and EANET has observations of $PM_{2.5}$, $PM_{10}$, $O_3$, CO, $SO_2$, $NH_3$, $NO_2$, $SO_4^{2-}$, $NO_3^-$, and $NH_4^+$ at more than 30 sites over East Asia countries (Dong and Fu, 2015b, a), so these two datasets are used for model evaluation in EAS. AERONET has AOD

measurements at more than 1,400 sites with a global coverage (Dubovik et al., 2000), so the AERONET data is categorized into EUR, RBU, and EAS region first and then applied for the model evaluation at the corresponding region. As some of the sites may not have valid measurements during the simulation period, only those with valid data are used and their locations are shown in Fig.1. Satellite retrieved AOD is collected from the MODIS product with 0.25°×0.25° grid resolution to investigate the spatial distributions

and column densities of aerosol simulated by the participating models.

Monthly mean surface concentrations from participating models are sampled at their own model grid cells containing the observational sites, and the corresponding measurements are also averaged on monthly scale to facilitate the evaluation. As no valid data is found for surface measurements of air pollutants in the RBU region, and the monthly trend of surface $O_3$, $PM_{2.5}$, and $PM_{10}$ are shown for EUR and

EAS in Fig.2. The evaluation statistics including mean bias (Simpson et al.) and coefficient of determination ($R^2$) are indicated in Fig.2 for the model ensemble mean only, calculated as the average of all participating models at the coarsest grid resolution (2.8°×2.8°). Although the mass concentrations measurements of aerosol sub-species including sulfate ($SO_4^{2-}$), nitrate ($NO_3^-$), ammonium ($NH_4^+$), organic aerosols (OA) and gas-phase species such as CO, $NH_3$, $NO_2$, and $SO_2$ are also available at some of the EBAS and EANET

stations, the data coverage is very sparse in terms of both number of sites and sampling periods, so the evaluations of these species are not discussed here but presented in the supplementary material (Table S1). In general, all participating models successfully reproduce the seasonal cycle of $O_3$ in EUR and EAS. The



model ensemble mean shows MB of only 4.4 µg/m$^3$ as compared to the EBAS observation in EUR. Relatively large biases (8-15 µg/m$^3$) are indicated in warmer months (from Jun. to Sep.), but meanwhile the standard deviation of measurement (indicated by vertical error bars in Fig.2) is even larger (10-15 µg/m$^3$), indicating that the measured O$_3$ concentrations vary significantly among the EBAS sites in the same

model ensemble grid due to the coarse resolution. Temporal variation of O$_3$ is also simulated well in EAS, although the models all tend to underestimate the high peaks in spring (Mar.-Apr.) and low bottoms in summer (Jul.-Sep.)

        Simulations of surface PM$_{2.5}$ concentrations are consistent among the participating models except that GEOSCHEMADJOINT suggests larger temporal variation than the other models. In EUR, the model

ensemble mean shows MB as -4.6 µg/m$^3$ against EBAS measurements. The seasonal cycle of PM$_{2.5}$ is less prominent as that of O$_3$ as indicated by the observations, but the ensemble mean generally captures the monthly changes with R$^2$ as 0.7. Underestimation of surface PM$_{2.5}$ concentration in EUR might be due to the fact that some of the measurements are affected by the local sources. As demonstrated in Fig.2(b), PM$_{2.5}$ are available from five EBAS stations. By examining their locations, we find that one of the stations is close

to highway (49.90°N, 4.63°E) and shows significantly higher PM$_{2.5}$ measurements than the others, which shall be attributed to the influences from not only the traffic emissions but also the wild fires from the forest nearby. These local impacts can hardly be captured by global models due to their coarse grid resolution. In EAS region, model ensemble mean shows a small MB as -1.6 µg/m$^3$ but poor correlation with measurement as the R$^2$ is only 0.2. The monthly dynamics of PM$_{2.5}$ is more prominent in EAS as that in EUR and the

models tend to miss the high peaks in spring (Apr.-May). As the anthropogenic emission in Asia is developed with top-town method, the predefined temporal profile applied during the modeling have been demonstrated to affect the model's capability of reproducing the temporal changes of PM$_{2.5}$ (Dong and Fu, 2015a). Simulation of PM$_{10}$ concentration shows good agreement between the model ensemble mean and the measurements in EUR, with MB of only -0.7 µg/m$^3$. The models systematically underestimate surface

PM$_{10}$ by -30.7 µg/m$^3$ in EAS but successfully reproduce the seasonal cycle. This is likely due to the fact that majority of the API and EANET stations are located in the urban area and thus get frequently affected by the local sources. Previous studies (Dong et al., 2015a) also suggested that the anthropogenic emission of primary PM$_{10}$ might be underestimated in China and subsequently lead to negative MB.

30        As no surface measurement of air pollutants is available in the RBU region, we evaluate the model simulated AOD against AERONET measurement and MODIS satellite product on monthly scale in all the three regions as shown in Fig.3. Most of the models fall into the two-fold range at both AERONET stations and MODIS grid cells. The participating models tend to overestimate AOD in the EUR region as compared to the AERONET observation, as the model ensemble mean shows MB of 0.1 and R$^2$ of 0.3. In the RBU

region, the model ensemble mean shows MB of only 0.05 yet the R$^2$ is only 0.2, indicating that there is a large discrepancy between model simulation and AERONET in terms of the temporal changes of AOD. The model ensemble mean shows best performance in EAS among all the three regions with MB of 0.1 and R$^2$ of 0.6, suggesting that models have good agreement with AERONET observation for both the level and the seasonal cycle of AOD. The simulated AOD are generally consistent between models, except that

CHASER is always 1-2 times higher than the others. The validations against MODIS product suggest slightly better performance of the models, as the model ensemble mean shows R$^2$ values as 0.5, 0.4, and 0.6 in EUR, RBU, and EAS respectively. In contrast to the overall overestimation indicated by evaluation against AERONET, the evaluation against MODIS suggests models tend to slightly underestimate the AOD



in all three regions with MB of -0.02, -0.04, and -0.03 in the EUR, RBU, and EAS regions respectively. This shall be due to the fact that AERONET has limited number of stations – there are 73, 11, and 15 stations in the EUR, RBU, and EAS regions respectively that have valid observations covering the simulation period – while MODIS has more comparable grid cells over the study domain.

The discrepancy between AERONET observations and MODIS product indicates that limited number of surface observations may not be sufficient to judge the overall performance of model since there is a high chance that observation may get affected by the local sources and subsequently biasing the assessment. To achieve a more solid understanding of the model performance, spatial distributions of the

simulated AOD from all participating models and the MODIS product are compared as shown in Fig.4. The Aerosol Comparisons between Observations and Models (AEROCOM) project has conducted a thoroughly evaluation of 14 global models and suggested the simulated AOD is in a two-fold range of the observations with mean normalized bias (MNB) varied between -44% and 27% (Huneeus et al., 2011). As presented in Fig.4, the model ensemble mean in this study shows good agreement with the MODIS production in terms

of spatial distribution, and the MNB values are 9.3%, 18.1%, and 44.9% in the EUR, RBU, and EAS regions respectively. These evaluation statistics are consistent with the evaluations by AEROCOM. But we also find some exceptions as CHASER significantly overestimate the AOD in China especially over the central and east coastal areas, indicating that the simulation bias may be generated by the model's treatment of the intensive anthropogenic emission over these areas. The SPRINTARS is also found to significantly

overestimate AOD over the Taklamakan Desert area, indicating that the bias shall be attributed to the model's capability of predicting wind-blown dust.

## 3. Result and Discussion

### 3.1 Seasonality of long-range transport impacts at surface layer

We start evaluating the long-range transport of $PM_{2.5}$ from the EUR and RBU source regions to the EAS receptor region by estimating the surface $PM_{2.5}$ concentration response on domain average scale under the emission perturbation scenarios. PM response ($\Delta PM$) is defined as the concentrations difference between the baseline scenario and the perturbation scenarios as:

$$\Delta PM_{EURALL} = PM_{BASE} - PM_{EURALL}$$

$$\Delta PM_{RBUALL} = PM_{BASE} - PM_{RBUALL}$$

To also understand the responses of aerosol sub-species, simulations of $SO_4^{2-}$, $NO_3^-$, $NH_4^+$, OA, and black carbon (BC) are collected from each of the participating models if it is available. Dust and sea salt are not analyzed in this study because emission perturbations are performed for anthropogenic sectors only. So in this study we assume that $\Delta PM_{2.5} = \Delta SO_4^{2-} + \Delta OA + \Delta BC + \Delta NO_3^- + \Delta NH_4^+$. For those models

reporting organic carbon (OC) instead of OA, an OC-to-OA conversion factor as 1.8 is applied to estimate OA following the method discussed in Stjern et al. (2016). For those models reporting only some of the sub-species and total $PM_{2.5}$, an extra species "other" is defined as subtracting the available sub-species from $PM_{2.5}$. For example, GEOS5 and SPRINTARS report mass concentrations of $SO_4^{2-}$, OA, BC, and $PM_{2.5}$, then for these two models we use: Other = $PM_{2.5}$ – ($SO_4^{2-}$ + OA + BC). Note that the CAM-chem model



reports sub-species for all scenarios but $NO_3^-$ for BASE scenario only, so no ΔOther is estimated for this model.

Long-range transport impacts from the EUR region are presented in Fig.5. Large variations of the simulated $PM_{2.5}$ responses are found among the models. The largest estimation of $\Delta PM_{2.5}$ is 0.16 µg/m³ estimated by GEOS5 in March, and the smallest $\Delta PM_{2.5}$ is 0.01µg/m³ estimated by EMEP in July. Regarding the seasonal cycle, majority of the models suggest the long-range transport has higher impact in winter and spring and lower impact in summer, well consistent with the $O_3$ long-range transport seasonality reported by the HTAP1 assessment (Dentener et al., 2010). CAM-chem does suggest higher values of $\Delta SO_4^{2-} + \Delta OA + \Delta BC + \Delta NH_4^+$ in July, but the seasonal cycle of $\Delta PM_{2.5}$ is unknown because $\Delta NO_3^-$ is not available. For most of the participating models, $\Delta SO_4^{2-}$ and/or ΔOA make larger contributions to $\Delta PM_{2.5}$ and show more prominent monthly changes than other sub-species. CAM-chem and GEOSCHEMADJOINT simulated $\Delta SO_4^{2-}$ shows monthly variations with a factor of 5, and GEOS5 suggests the monthly dynamics of ΔOA is with a factor of 8. The model ensemble mean suggests that the largest long-range transport impact of $\Delta PM_{2.5}$ is 0.064 µg/m³ in March and the smallest impact is 0.035 µg/m³ in September, and the contributions from ΔBC, $\Delta SO_4^{2-}$, ΔOA, $\Delta NO_3^-$, and $\Delta NH_4^+$ are 3%, 45%, 19%, 17%, and 16% respectively.

Long-range transport from the RBU to the EAS region is presented in Fig.6. The highest $\Delta PM_{2.5}$ is estimated by GEOS5 as 0.19 µg/m³ in March, while the lowest ΔPM is indicated by GEOSCHEMADJOINT as 0.018 µg/m³ in July. Similar to the response under EURALL scenario, long-range transport from the RBU region is also substantially contributed by $\Delta SO_4^{2-}$, but $\Delta NO_3^-$ and $\Delta NH_4^+$ share more significant portions in $\Delta PM_{2.5}$. Most of the models suggest relatively lower values of ΔOA except for GEOS5, which suggests up to 0.1µg/m³ ΔOA in March. The model ensemble mean suggests maxima of $\Delta PM_{2.5}$ as 0.101µg/m³ in March and the minima as 0.065µg/m³ in August, and the contributions from ΔBC, $\Delta SO_4^{2-}$, ΔOA, $\Delta NO_3^-$, and $\Delta NH_4^+$ are 2%, 43%, 14%, 20%, and 21% respectively. Percentage contributions are generally less than 3%, yet the highest contributions could be up to 3-4% for $\Delta SO_4^{2-}$, $\Delta NO_3^-$, and $\Delta NH_4^+$ as suggested by EMEP. The relatively lower contribution of ΔOA and higher contributions of $\Delta NO_3^-$ and $\Delta NH_4^+$ under the RBUALL scenario is probably due to the low temperature in the RBU source region, which may extend the lifetime of gas-phase precursors ($SO_2$, $NO_x$, and $NH_3$) and enhance the export of secondary inorganic aerosols produced during the journey of long-range transport. In fact, the low temperature also favors SOA production from VOC due to the partitioning to the condensed phase. CAM-chem suggests the contribution of ΔSOA in ΔOA is 32% under the RBUALL scenario and 28% under the EURALL scenario, and model ensemble mean also shows that more OA is transported from RBU (0.01µg/m³) than that from EUR (0.008µg/m³), although the anthropogenic NMVOC and OC emission from EUR is 10% and 70% higher respectively. But the low temperature seems affect the $SO_2$, $NO_x$, and $NH_3$ more by influencing the chemical kinetics and slow down the production of PM at the source region, which may allow more uplift motion of the gas-phase precursors, and finally result in more $\Delta SO_4^{2-}$, $\Delta NO_3^-$, and $\Delta NH_4^+$ produced during the long-range transport pathway. More research effort is necessary to explicitly understand the export of precursors and secondary inorganic aerosols traveling from high latitude areas.





## 3.2 Long-range transport above and within the PBL

The HTAP phase 1 (HTAP1) report suggests that long-range transport of air pollutants from Europe to Asia are identified at two major different heights: within and above 3km respectively, and the upper path is believed to be more important due to the existence of the Westerlies, especially when the emission source area is close to the jet stream. While the modeling effort (Eckhardt et al., 2003;Stohl et al., 2002) referenced by the HTAP1 report is mainly investigating the influence of North Atlantic Oscillation (NAO) on air pollutants transport towards the Arctic, the Europe to Asia transport pathways are identified based on spatial distributions of simulated CO column density, and the contributions from upper and lower levels transport remain unknown. The transport pathways above and within 3km are commonly used by previous studies in order to distinguish the long-range transport above and within the free troposphere, but 3km was apparently a rough estimation of the PBL height. Although PBL transport plays a dominant role in air pollutants dispersion at local scale, the intensity of long-range transport exclusively within the PBL is believed to be negligible because it is frequently affected by the land surface, turbulence, and exchange with the free troposphere. The transport from Europe to Asia estimated with model experiment in this study however, may also exist within the PBL since the emission perturbation is performed on continental scale, and there is a large portion of remote areas with flat topography in the Asia-Stan region laying between Europe and East Asia. As very limited modeling efforts have been devoted to investigate the transport within the PBL, we compare the amount of PM responses within and above the PBL in this study to examine the contributions of long-range transport in different atmosphere layers. Annual average PBL height is about 1.5km (880hPa-850hPa) above surface ground over our study domain on annual average scale, and instead of assuming a constant PBL height, we use the monthly PBL data from the SPRINTARS model because it is the only one that uploads. To enable the comparison of PM transported within and above the PBL, we use the column density instead of mass concentration, defined as below:

$$\Delta PM_{within} = \sum_{layer=surface\ layer}^{PBL} \Delta PMC_{layer} \times HT_{layer}$$

$$\Delta PM_{above} = \sum_{layer=PBL+1}^{model\ top} \Delta PMC_{layer} \times HT_{layer}$$

where $\Delta PM_{above}$ ($\Delta PM_{within}$) is the $\Delta PM$ transported above (within) the PBL, $\Delta PMC$ is the mass concentrations responses under the perturbation scenarios at each layer, and HT is the model layer thickness. Fig.7 presents the spatial distributions of model simulated $\Delta PM_{within}$ and $\Delta PM_{above}$ under the EURALL scenario, as well as the longitude-pressure cross sections of $\Delta PMC$ estimated by the participating models. It is important to note that PM mentioned in this section refers to the lump sum of $SO_4^{2-}$, OA, and BC (because these are the sub-species available from all participating models) to enable the inter-comparison between the models.

Transport from the EUR to the EAS region shows generally consistent spatial distributions by all participating models. Long-range transport of PM above the PBL is mainly distributed along 40ºN and higher latitude, where the impact can reach even further towards the west Pacific. The lower latitude (30ºN-40ºN) transport of PM is blocked by the Pamirs, Tianshan, and Altay Mountains due to the elevated topography along the western boundary of China. The long-range transport of PM within PBL is mostly blocked shortly after exported from Europe at the eastern side of Black Sea along Iran, Georgia, and



Armenia, while the rest of it travels along 45ºN and above latitudes towards East Asia. All participating models suggest that PM is firstly carried from EUR towards northeast direction over Siberia, Mongolia and Northeast part of China, and then down to lower latitude areas over North China Plain (NCP). This transport pathway is well consistent with the findings by the HTAP1 assessment (Dentener et al., 2010). $\Delta PM_{above}$ is

found substantially higher than $\Delta PM_{within}$ over the EAS receptor region. Large values of $\Delta PM_{above}$ suggest that the long-range transport may also play an important role in affecting the shortwave radiative forcing budget since the aerosol may suspend above the cloud. Deposition of PM from upper air down to the surface layer may also subsequently affect to the near surface layer air quality. Most models show gradually decreased $\Delta PM_{above}$ and $\Delta PM_{within}$ from EUR to EAS, but SPRINTARS shows non-negligible PM changes

along the southeast coast of China, which could be due to the production of secondary $SO_4^{2-}$ converted from long-range transport $SO_2$, discussed earlier in section3.1. The largest long-range transport impact is estimated by CHASER and smallest impact is estimated by EMEP, but no significant diversities are found among the models regarding the intensity of $\Delta PM_{above}$ and $\Delta PM_{within}$. The longitude-pressure cross sections of the PM responses present a clear depict of the long-range transport from EUR to EAS at different height.

The PM responses along the longitude can reach up to higher than 500hPa over the EUR region (10ºE-40ºE), indicating a significant uplift motion of the air pollutants over Europe. Majority of the eastward transport PM is blocked at 45ºE-50ºE due to the elevated topography. In the upper layer above 800hPa however, PM is slightly less affected by the topography and can transport further towards the EAS region, where it deposits to near surface layer subsequently. Both the spatial distributions of $\Delta PM_{within}$ and the cross

sections of $\Delta PMC$ suggested that the inter-continental transport of aerosol does occur within PBL, although the intensity is less significant as compared to that above PBL. Under the ERUALL scenario, $\Delta PM_{within}$ contribution to the total column density of $\Delta PM$ is 34% estimated by the model ensemble mean, with the lowest contribution estimated by EMEP as 22% and highest contribution estimated by GEOSCHEMADJOINT as 38%.

Long-range transport from RBU follows the similar pathway as that from EUR to EAS, as shown in Fig.8, which is likely because most of the RBU anthropogenic emissions are located at the European part of Russia and Ukraine. PM responses are also relatively more significant in the upper air above PBL, which spread along 45ºN and higher latitude and affect the north part of China, North Korea, South Korea, and

Japan. Long-range transport from RBU is slightly larger than that from EUR for both above and within the PBL. Spatial distributions of $\Delta PM_{above}$ and $\Delta PM_{within}$ suggest that RBU exported air pollutants can travel further towards the west Pacific. Cross sections of PM concentrations suggest that RBU emitted PM shows a much lower plume rise height in the source region as compared to that over EUR. PM response under the RBUALL scenario is also found to exist at up to 500hPa in the source region, but majority of plume is

within 800hPa.

## 3.3 Trend and inter-annual variability of the long-range transport

The global anthropogenic emissions have changed significantly especially over East Asia during the past decade, thus the long-range transport impact and its relative importance may have also changed as

well. In this section, we compare the impact estimated in this study for the year 2010 with the assessment reported by HTAP1 for the year 2000 to reveal the trend of inter-continental transport. We also analyze the





HTAP2 simulations for the year 2008 and 2009 to probe into the inter-annual variability of the long-range transport. To properly interpret the HTAP1 report and the HTAP2 modeling results, it is important to realize that the regions definitions are moderately different between the two modeling experiments. HTAP1 used straight latitude and longitude boundaries to define the domain coverage of each region (Fiore et al., 2009),

while HTAP2 applies national boundaries (one exception in the Northern Hemisphere is the Arctic region, defined as being North of 66°N latitude), thus the spatial coverage of "EU" (25°N-65°N; 10°W-50°E) defined by HTAP1 is slightly different from "EUR" defined by HTAP2, although both of them represent the European region. A similar discrepancy exist for the definition of East Asia between the two experiments, as the HTAP1 defined "EA" (15°N-50°N; 95°E-160°E) is smaller than the EAS region with

less coverage on the west and north side of China. Consequently, when referring to "long-range transport from Europe to East Asia", neither the source (Europe) nor the receptor region (East Asia) share exactly the same meaning between HTAP1 and HTAP2. These different region definitions will determine how to interpret the modeling results as will be discussed later in this section. In addition, emission perturbations in source regions performed in both HTAP1 and HTAP2 experiments are 20% instead of 100%, thus the

full contributions from the EUR or RBU to the EAS region remain unknown. Although the PM response is not exactly proportional to emission perturbation, previous studies (Leibensperger et al., 2011;Liu et al., 2008) suggested that it is reasonable to linearly extrapolate it when evaluating the inter-continental source-receptor relationship because the non-linear relationship between precursor emission changes and PM responses is only effective locally. The HTAP1 assessment reported that surface $SO_4^{2-}$ concentrations is

reduced by 12%-14% from 20% local emission reduction in East Asia, Europe, and North America, corresponding to 60%-70% reduction under 100% local emission reduction if the responses are extrapolated linearly. Yet model experiments show that the real 100% emission perturbation simulations suggest 80-82% surface $SO_4^{2-}$ concentrations reduction due to "oxidant limitation" over these polluted areas. However, this relationship becomes linear during trans-oceanic transport due to the relatively short lifetime of precursors

as compared to the travel duration. So in this study, we use the $Full\_Impact$ to represent the PM responses from 100% emission perturbation at EUR and RBU by scaling the PM responses under the 20% emission perturbation conditions by a factor of 5, which provide a rough but direct estimation of the full contributions of long-range transport:

$$Full\_Impact_{EUR} = 5 \times \Delta PM_{EUR}$$

$$Full\_Impact_{RBU} = 5 \times \Delta PM_{RBU}$$

and:

$$Full\_Impact_{EUR\%} = \frac{Full\_Impact_{EUR}}{PM_{BASE}} \times 100\%$$

$$Full\_Impact_{RBU\%} = \frac{Full\_Impact_{RBU}}{PM_{BASE}} \times 100\%$$

In addition, we also defined the $Relative\_Impact$ in this study to represent the relative importance

of long-range transport in contrast to the local emission, as the ratio of PM responses under 20% emission perturbation in source region (i.e. EUR, RBU) to the PM responses under 20% emission perturbation in the receptor region (i.e. EAS):

$$Relative\_Impact_{EUR\%} = \frac{\Delta PM_{EUR}}{\Delta PM_{EAS}} \times 100\%$$



$$Relative\_Impact_{RBU\%} = \frac{\Delta PM_{RBU}}{\Delta PM_{EAS}} \times 100\%$$

Full impact and relative impact are calculated with model ensemble mean to represent the averages, and with individual modeling results to estimate the minima and maxima, as summarized in Table 2. The HTAP1 experiment only reported the assessment of $SO_4^{2-}$, BC and OA, so this section will focus on the

analysis and comparison of these species. As mentioned earlier, the EAS region is different from the EA region defined in HTAP1, so we also calculate the full impact and relative impact for the EA region but with HTAP2 modeling data to enable the comparison between the assessments from the two experiments. We first compare the 2000 EU impact on EA with the 2010 EUR impact on EA. The long-range transport shows prominent decreasing trend for all investigated species as shown in Table2. The full impact of Europe

long-range transport on surface $SO_4^{2-}$ concentration decreased from 0.15µg/m³ (5.0%) in 2000 to 0.02µg/m³ (0.5%) in 2010, which shall be due to the significant reduction of $SO_2$ anthropogenic emission in Europe from 9.95Tg in 2000 to 6.18Tg in 2010 (anthropogenic emissions are summarized in Table S2). The full impacts of Europe long-range transport on surface BC and OA are also found to decrease by a factor of 2-5 for both absolute concentrations and percentage contributions during the 10 years period. Anthropogenic

emissions of BC, OC, NMVOC, and primary PM in Europe are decreased by 21%, 4%, 37%, and 2% respectively and their emissions in East Asia are increased by 39%, 21%, 38%, and 32% respectively from 2000 to 2010. The emission increase in East Asia shall be response for the enhanced surface PM concentrations simulated under the baseline scenario, and the emission reductions in EUR are consistent with the decreasing trend of the long-range transport contributions estimated by the models.

We then investigate the inter-annual variability of the long-range transport by examining the EUR to EAS and the RBU to EAS impact from 2008 to 2010. The model estimated $Full\ Impact_{EUR\%}$ shows moderate changes by 15%-30% for all species from year to year, with no significant trend is found. The $Full\ Impact_{RBU\%}$ shows relatively larger inter-annual changes. As the anthropogenic emissions from the RBU region has steadily decreased by ~9% from 2008 to 2010, the large dynamics of $Full\ Impact_{RBU\%}$ is

more likely due to the fact that only one model (CAM-chem) is available to estimate the RBU impact in 2008 and 2009 and thus the assessment may be biased. While the estimation for 2010 is calculated with multi-model ensemble mean, the estimations for the other two years are determined by one model only and need to be further validated.

We finally analyze the relative importance of long-range transport. The HTAP1 reported that the

overall contribution to $SO_4^{2-}$ and OA from EU to EA is 2.9% in 2000, and in this study the estimated relative impact in 2010 is 2.2%, indicating that long-range transport is playing a less important role as compared to the local anthropogenic emission in terms of affecting the surface air quality in East Asia. In contrast, 20% anthropogenic emission reductions in the EAS region lead to surface concentration of $SO_4^{2-}$+OA decreased by 16.8% in 2000 and 14.1% in 2010, suggesting that the non-linear relationship between precursor and

PM becomes more significant when the anthropogenic emissions increase. It also indicates that to achieve a better air quality with lower PM concentrations, more efforts shall be devoted to reduce the emissions in 2010 because the top 20% emission reduction would lead to less PM response as compared to that in 2000.

## 3.4 Long-range transport impact during the haze episode

As the annual average full impact for aerosol-sub species are presented in last section, in this section we evaluate the full impact during the haze episodes for $PM_{2.5}$. We first use the National Climate Data Center (NCDC) observations to identify the locations and periods of haze in China, and then analyze the





long-range transport impacts during these identified haze episodes. Haze is defined as visibility less than 10km and relative humidity less than 90% (Fu et al., 2014). As most of the haze (locations of NCDC sites and full map of haze shown in Fig.S1) are located over central and eastern part of China (CEC), in this section we focus the analysis of long-range transport impacts on the CEC subdomain (20°N-55°N; 100°E-135°E). The full impacts during the haze episodes (HAZE) are estimated and compared with the full impacts throughout the year of 2010 (AAVG) as shown in Table 3.

CAM-chem and GEOS5 has no daily surface data available so data from the rest 4 participating models are analyzed in this section. The models suggest that the $PM_{2.5}$ baseline concentrations during haze episodes are substantially higher than the annual averages, with the largest difference between HAZE and AAVG is estimated by CHAER as 27.27µg/m³ and the smallest difference estimated by GEOSCHEMADJOINT as 2.56µg/m³. The full impacts of long-range transport from the source regions are also higher during the haze episodes by a factor of 2-3 than the annual averages. As estimated by the model ensemble mean, on annual average scale the $Full\_Impact_{EUR}$ is 0.35µg/m³, contributing to 1.7% of the surface $PM_{2.5}$ in the EAS region. During the haze episode however, $Full\_Impact_{EUR}$ is 0.99µg/m³, contributing to 3.1% of the surface $PM_{2.5}$ in the ECE region. The impact from the RBU region is also found more significant during haze episodes, as the $Full\_Impact_{RBU}$ increased from 0.53µg/m³ (2.6%) to 1.32µg/m³ (4.1%). Higher values of $Full\_Impact_{EUR}$ and $Full\_Impact_{RBU}$ suggest that more fine particles are transported from the EUR and RBU source regions when China is suffering from haze.

As shown in Fig.9. The spatial distributions of the long-range transport full impacts during the haze episodes demonstrate a very similar pattern among the participating models. The $Full\_Impact_{EUR\%}$ is most significant over the northeast corner of China, and gradually decreases towards the southeast direction. The intensity of $Full\_Impact_{EUR\%}$ estimated by models however, show large difference as the maxima estimated by SPRINTARS is 10.5% and the minima estimated by EMEP is 0.4%. The spatial distributions of $Full\_Impact_{RBU\%}$ shown by the models are similar to that of $Full\_Impact_{RBU\%}$, but the intensity of long-range transport from the RBU region is generally larger. The numbers presented in Table 3 have demonstrated the general full impacts during all haze episodes, but we are still unaware of how those individual haze episodes are affected by the long-range transport. So we also summarize the histograms of daily full impacts during the haze episodes. The frequency of the histogram is calculated as:

$$Frequency_{Full\_Impact=i\%} = \frac{\#HazeEvent_{i\%}}{\sum_{i=1}^{MaxFI=15} \#HazeEvent_{i\%}} \times 100\%$$

and it satisfies:

$$\sum_{i=1}^{MaxFI=15} Frequency_{Full\_Impact=i\%} = 100\%$$

We define $MaxFI = 15$ to represent the upper boundary as $Full\_Impact \geq 15\%$. This value (i.e. 15%) contribution is selected in order to compare the full impact from long-range transport against the $PM_{2.5}$ response under 20% local emission control in the EAS region. As shown in Table 2, surface concentration of $SO_4^{2-}$+OA is reduced by ~15% under the EASALL scenario. So if $Full\_Impact_{EUR} \geq 15\%$, it indicates that the long-range transport from EUR may have an equivalent or even more significant contribution to the surface $PM_{2.5}$ as that produced from 20% of the local anthropogenic emission. We define $\#HazeEvent_{i\%}$ as the number of haze events that satisfies: $(i-1)\% < Full\_Impact \leq i\%$ and is calculated as:



$$HazeEvent_{i\%} = \sum_{d=1}^{365} H_{d,r,c}$$

$H_{d,r,c}$ is the haze event at day $d$, row $r$, and column $c$, defined as:

$$H_{d,r,c} = \begin{cases} 1, if\ RH_{d,r,c} < 90\%\ and\ visibility_{d,r,c} < 10km, and\ i\% < Full\_Impact_{d,r,c} \leq (i+1)\% \\ 0, otherwise \end{cases}$$

So with $Frequency_{Full\_Impact=i\%}$, we can estimate the percentage of the haze episodes during which the
long-range transport contributes to $i\%$ of the surface PM$_{2.5}$. The values of $Frequency_{Full\_Impact=15\%}$ are
indicated in the histogram plots as shown in Fig.9. The SPRINTARS estimated $Frequency_{Full\_Impact=15\%}$
is 5.5%, suggesting that during almost 5.5% of the haze episodes in China, long-range transport from
Europe contributed to at least the equivalent amount of surface PM$_{2.5}$ concentration as that generated from
20% of local anthropogenic emission, while the other models' estimations range from 0.01% to 1.9%. The
participating models suggest that $Frequency_{Full\_Impact=15\%}$ ranges from 0.1% to 5.5% with the model
ensemble mean estimates as 1.8%. The influence from the RBU region shows slightly higher value of
$Frequency_{Full\_Impact=15\%}$ as the model ensemble mean estimates as 2.2%. Although significant
variations are found among the model estimations, all participating models suggest that non-negligible
values of $Frequency_{Full\_Impact=15\%}$ and $Frequency_{Full\_Impact=15\%}$, indicating the important
contributions of long-range transport to haze episodes in China.

Although the high surface PM$_{2.5}$ is believed to be the most direct reason for causing haze condition,
visibility cannot be represented by PM$_{2.5}$ mass concentration only since it is also determined by the optical
properties, number concentrations, and size distributions of the aerosols. Thus the analysis of PM
concentration response depicts only partially of the impact of long-range transport during haze episodes.
Calculating model predicted visibility requires the detailed aerosol information mentioned above which is
not available from any of the participating models. So we use the Koschmieder equation (Han et al., 2013)to
estimate the model simulated visibility from aerosol extinction coefficient ($\beta$) as:

$$Visibility = \frac{3.912}{\beta}$$

Modeled visibility is calculated for SPRINTARS only since the other participating models has no
surface layer extinction coefficient available. The long-range transport impact on visibility change and
number of haze days change are shown in Fig.10. It shall be noticed that SPRINTARS estimated long-range
transport impact of surface PM$_{2.5}$ is the highest among the participating models, thus the analysis of
visibility change shown in Fig.10 shall represent the upper boundary of model estimations. The spatial
distribution of visibility changes agree well with the distribution of surface PM$_{2.5}$ responses. Visibility is
reduced by up to 10km along the northeast boundary of China, which is likely due to the fact that these
areas receive the most significant amount of the long-range transport aerosols from the EUR and RBU
regions. The number of haze days changes however, are mostly distributed in the NCP and along the east
coast of China. The long-range transport results in 1-3 days of extra haze over these areas throughout the
year. The total number of haze events ($\sum_{i=1}^{MaxFI=15} \#HazeEvent_{i\%}$) estimated by the SPRINTARS model
is 18566, 18538, and 18546 under the BASE, EURALL, and RBUALL scenarios, suggesting that that





transport from the EUR and RBU region contribute to an extra of 0.15% and 0.11% haze events respectively.

## 4. Summary and conclusions

To estimate the long-range transport contributions to the surface aerosol concentrations in East Asia, this analysis uses 6 global models participating in the HTAP2 experiment. Simulations for the year 2010 from baseline scenario and 20% anthropogenic emission perturbation scenarios are explored to estimate the long-range transport from the Europe and Russia/Belarussia/Ukraine source regions respectively. We find that on annual average scale, long-range transport from Europe contributes 0.04-0.06 $\mu g/m^3$ (0.2-0.8%) to the surface $PM_{2.5}$ concentration in East Asia as indicated by the 20% emission perturbation experiment, with majority of the transported aerosols are $SO_4^{2-}$ and OA at 43% and 19% respectively. Long-range transport from Russia/Belarussia/Ukraine shows slightly higher impact with contributions of 0.07-0.10$\mu g/m^3$ (0.3-0.9%) to the surface $PM_{2.5}$ in East Asia, within which the $NO_3^-$ and $NH_4^+$ responses share bigger slices as 20% and 21% respectively, larger than that of OA as 14%. As compared to the impact from Europe to East Asia, more secondary inorganic aerosols are transported from the Russia/Belarussia/Ukraine region despite the fact that the 2010 anthropogenic emission from RBU is 40-50% lower than that from EUR for $SO_2$, $NO_x$, and $NH_3$. Our analysis suggests that the lower temperature in RBU may result in extended lifetime of the gas-phase precursors, which are gradually converted to secondary inorganic aerosols during the transport pathway to East Asia, yet further modeling experiment is necessary to explicitly explore the temperature impact on long-range transport.

By investigating the PM responses in different atmosphere layers, we find that long-range transport exist both within and above the PBL, although the upper level transport takes a larger portion as 66% of the total PM column density response in East Asia. Spatial distributions of the PM responses suggest that the long-range transport from Europe and Russia/Belarussia/Ukraine are both predominantly blocked at western side of China due to the elevated topography of Pamirs, Tianshan, and Altay Mountains, where the rest of the exported pollutants are carried by the Westerlies along 45ºN and higher latitude towards China, North Korea, South Korea, Japan, and the west Pacific.

Comparison between the HTAP1 assessment and the estimation from this study reveals the 10 years decreasing trend of long-range transport from Europe to East Asia. When extrapolating the impact of 20% anthropogenic emission perturbation by a factor of 5 to estimate the full impact, contributions to surface concentrations are decreased from 5.0%, 1.0%, and 0.4% in 2000 to 0.5%, 0.2%, and 0.2% in 2010 for $SO_4^{2-}$, BC, and OA respectively. This comparison may contain uncertainty because of the different model ensemble compositions between HTAP1 and this study, but the trend of the long-range transport impacts from 2000 to 2010 found in this study was consistent with the implications from the emissions changes. The simultaneously emission reduction in Europe and emission enhancement in East Asia shall be responsible for the decreasing trend. The surface concentrations of $SO_4^{2-}$, BC, and OA in East Asia are also increased by 14%, 50%, and 140% from 2000 to 2010, well consistent with many of the local measurements reported in recent years. It is important to emphasize that based on the model ensemble mean estimations, despite the fact that baseline of 2010 anthropogenic emission is substantially higher (20-40%) than that in 2000, a same percentage reduction of the local anthropogenic emission will lead to less benefit in terms of reducing the ambient PM concentrations in the 2010 scenario, indicating the increasingly more difficulties for air quality management in East Asia.



The long-range transport impact during haze episodes in China are estimated by using the NCDC surface observations to identify the haze events, on top of which the HTAP2 experiments are analyzed to quantify the changes of surface $PM_{2.5}$, visibility, and number of haze days. Despite the significant discrepancy between the models, all participants demonstrate that the full impacts during haze episodes are

more significant than that on annual average scale. Estimations with the model ensemble mean suggest that the full impacts from EUR and RBU are $0.99\mu g/m^3$ (3.1%) and $1.32\mu g/m^3$ (4.1%) respectively during haze episodes, significantly higher than the annual averages. The model ensemble also suggest that during 5.5-5.7% of the haze episodes, long-range transport can contribute to surface $PM_{2.5}$ as much as that generated from 20% of local anthropogenic emission. Based on analysis with the SPRINTARS model output,

visibility is reduced by up to 10km with the largest impact found along northeast China, and the impact gradually decreases towards southeast and causes less than 500m visibility reduction. The enhancement of number of haze days however, is found mainly located at the North China Plain and southeast coast area of China, where most of the places receive extra 1-3 haze days due to the influence of long-range transport. We find that throughout the full year of 2010, number of haze event in our studying domain is increased by

0.15% and 0.11% due to the long-range transport from the Europe and Russia/Belarussia/Ukraine region respectively.

## 5. Acknowledgements:

This work was partly supported by Natural Science Foundation of China (41429501). We would like to thank the UN-ECE CLRTAP (EMEP), AMAP, and NILU for supporting the EBAS

database with air pollutants measurements. We would also like to acknowledge NOAA NCDC to provide the public accessible meteorology observations. We thank the Oak Ridge Leadership Computing Facility (OLCF) at Oak Ridge National Lab (ORNL) for providing computer sources.



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





Figure Captions

Figure 1. The HTAP2 source and receptor regions for EUR (green), RBU (red), and EAS (grey). Sites marked with the same symbols are from the same observation network: red circles represent API, blue squares represent
AERONET, orange diamonds represent EANET, and yellow triangles represent EBAS.

Figure 2. Monthly mean surface concentrations of $O_3$ (left column), $PM_{2.5}$ (center column), and $PM_{10}$ (right column) for the year 2010 in the EUR (upper row) and EAS[1] (lower row) regions from observations and model simulations[3]. Observations (bold black lines with vertical error bars) represent the averages of all sites falling within the same ensemble grid (bold red lines), and the vertical error bars[2] depict the standard deviation across the sites in the same
ensemble grid. Models are sampled at the nearest grid to each station, multiple stations within the same model grid are averaged to represent the paring observation.

Figure 3. Monthly average AOD comparison between the models and AERONET (upper row) and between the models and the MODIS (bottom row) in EUR (left column), RBU (center column), and EAS (right column). Models are represented by markers with different colors and styles. Evaluation statistics (MB and $R^2$) are indicated for model
ensemble mean in the upper left corner of the scatter plot. The solid black line is the 1:1 line whereas the black dash contours represent the 1:2 and 2:1 lines.

Figure 4. Spatial distributions of AOD from MODIS and model simulations. Evaluation statistics of each model are indicated at the lower left corner of the plot.

Figure 5. Monthly averages of surface aerosol response in the EAS receptor region under the EURALL scenario. Solid
bars with different colors represent the responses of different aerosol.

Figure 6. Same as Figure 5 but under the RBUALL scenario.

Figure 7. Annual averages of PM column density responses (calculated as $\Delta PM = \Delta BC + \Delta SO_4^{2-} + \Delta OA$) under the EURALL scenario within (left column) and above (middle column) PBL, and the corresponding longitude-pressure cross sections of PM concentrations (averaged over 10ºN-70ºN) estimated by participating models.

Figure 8. Same as Figure 7 but under the RUBALL scenario

Figure 9. Spatial distributions and histograms of the long-range transport full impacts during the haze episodes.

Figure 10. Reduction of visibility (left column) and enhancement of number of haze days (right column) under the EURALL (upper row) and RBUALL (lower row) scenarios.



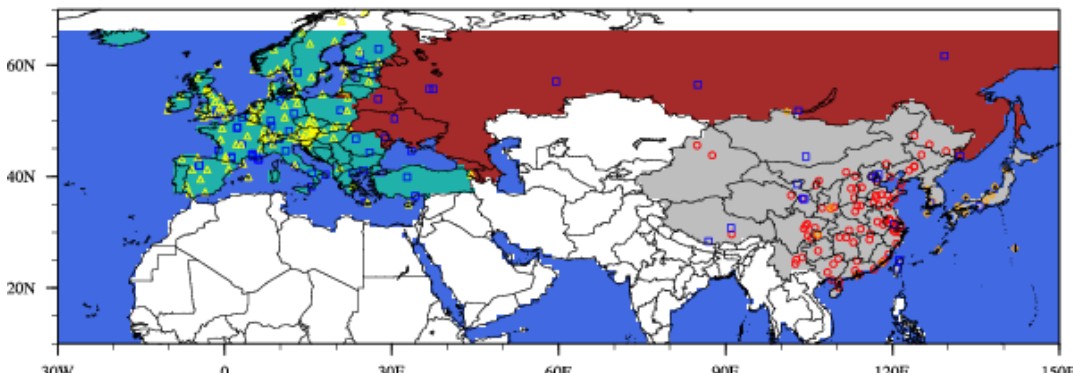

Figure 1. The HTAP2 source and receptor regions for EUR (green), RBU (red), and EAS (grey). Sites marked with the same symbols are from the same observation network: red circles represent API, blue squares represent AERONET, orange diamonds represent EANET, and yellow triangles represent EBAS.




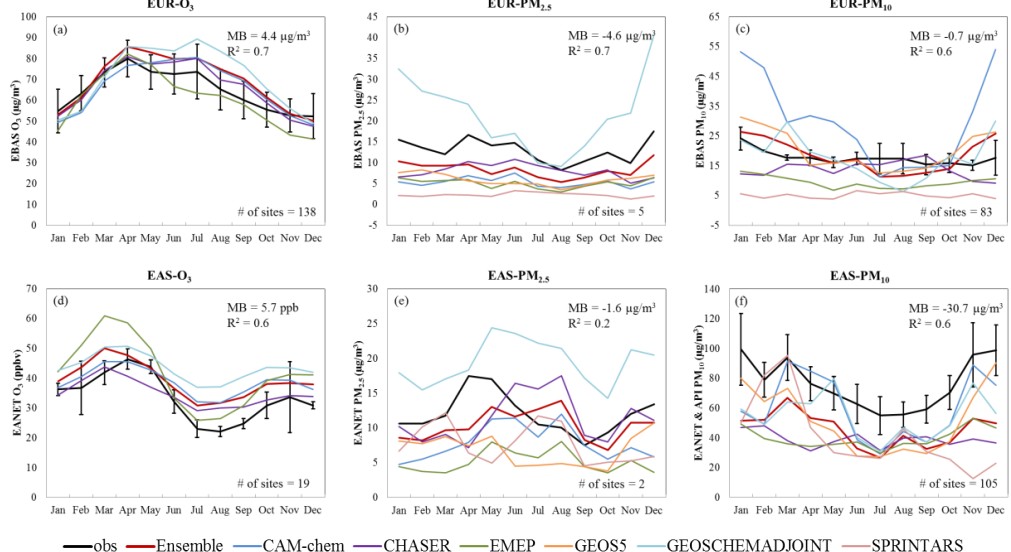

Figure 2. Monthly mean surface concentrations of $O_3$ (left column), $PM_{2.5}$ (center column), and $PM_{10}$ (right column) for the year 2010 in the EUR (upper row) and EAS[1] (lower row) regions from observations and model simulations[3]. Observations (bold black lines with vertical error bars) represent the averages of all sites falling within the same ensemble grid (bold red lines), and the vertical error bars[2] depict the standard deviation across the sites in the same ensemble grid. Models are sampled at the nearest grid to each station, multiple stations within the same model grid are averaged to represent the paring observation.

[1]$PM_{10}$ from API and EANET are used together to represent the observations in EAS region.

[2]$PM_{2.5}$ observations in EUR and EAS region have no standard deviation because there are no sites with valid measurements fall into the same model ensemble mean grid.

[3]Most participating models report the $PM_{2.5}$ mass concentration except that CAM-chem only reports the aerosol sub-species, so we calculate the CAM-chem simulated $PM_{2.5}$ by following the formula described in Silva et al. (2013).





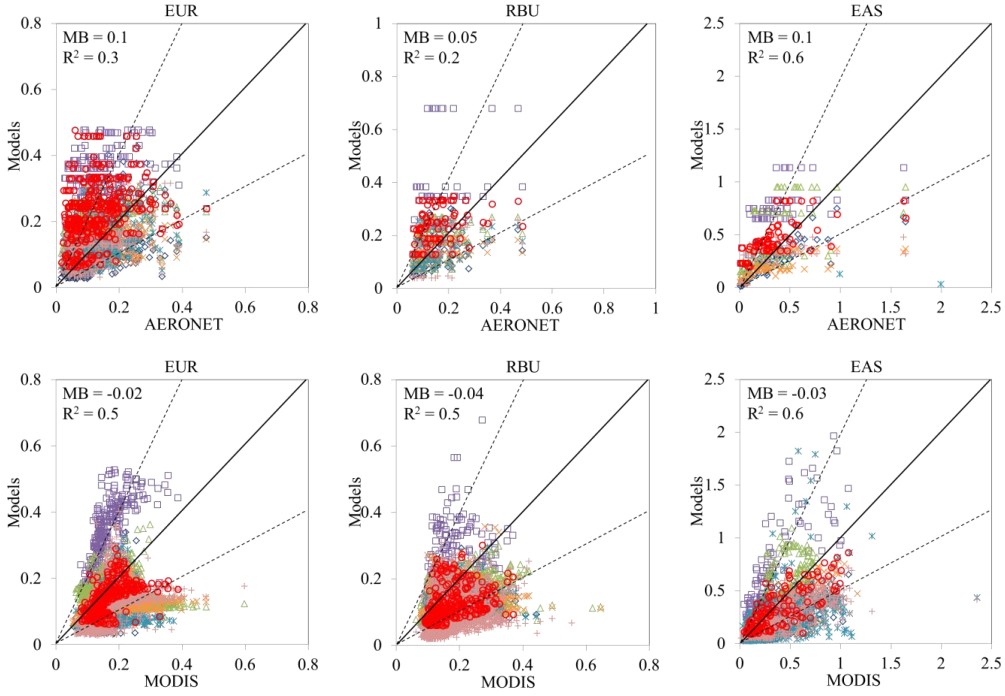

○ Ensemble ◇ CAM-chem □ CHASER △ EMEP ✕ GEOS5 ✹ GEOSCHEMADJOINT + SPRINTARS

Figure 3. Monthly average AOD comparison between the models and AERONET (upper row) and between the models and the MODIS (bottom row) in EUR (left column), RBU (center column), and EAS (right column). Models are represented by markers with different colors and styles. Evaluation statistics (MB and $R^2$) are indicated for model ensemble mean in the upper left corner of the scatter plot. The solid black line is the 1:1 line whereas the black dash contours represent the 1:2 and 2:1 lines.





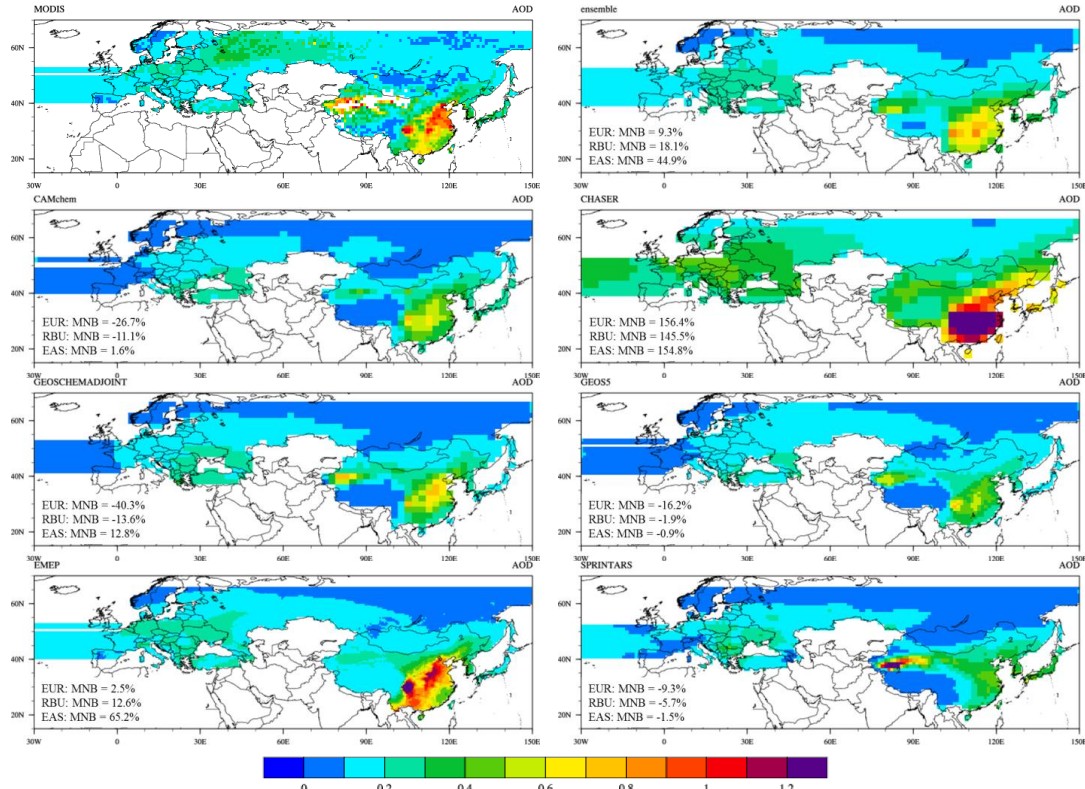

Figure 4. Spatial distributions of AOD from MODIS and model simulations. Evaluation statistics of each model are indicated at the lower left corner of the plot.





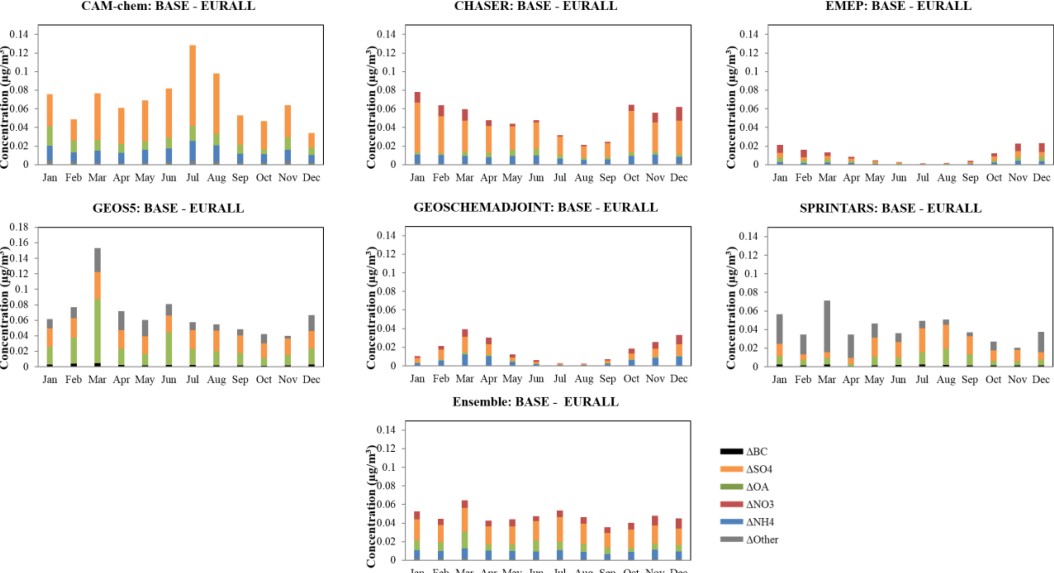

Figure 5. Monthly averages of surface aerosol response in the EAS receptor region under the EURALL scenario. Solid bars with different colors represent the responses of different aerosol.



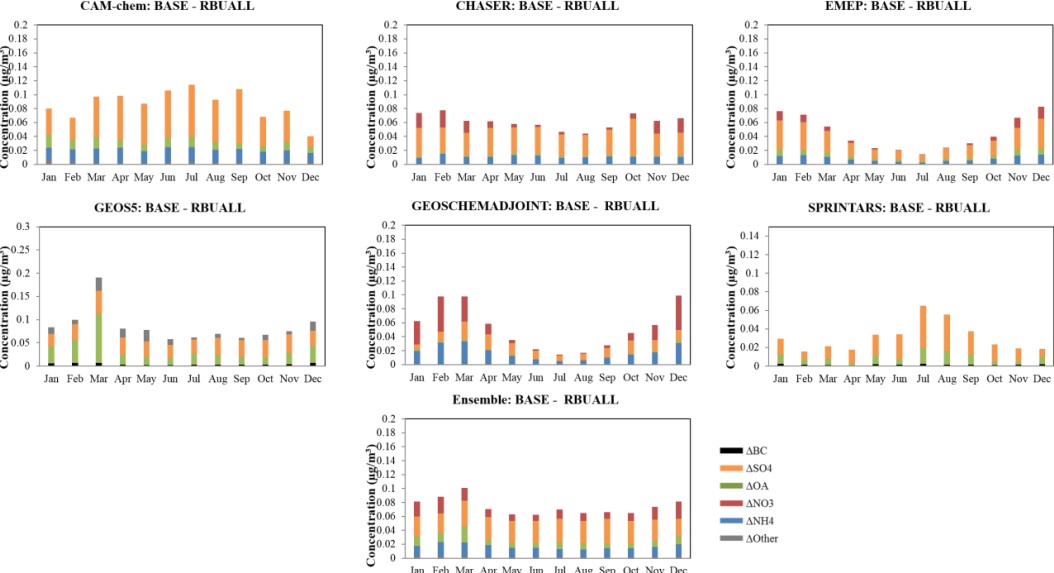

Figure 6. Same as Figure xxx but under the RBUALL scenario.







Figure 7. Annual averages of PM column density responses (calculated as ΔPM=ΔBC + ΔSO₄²⁻ + ΔOA) under the EURALL scenario within (left column) and above (middle column) PBL, and the corresponding longitude-pressure cross sections of PM concentrations (averaged over 10°N-70°N) estimated by participating models.





Figure 8. Same as Figure 7 but under the RUBALL scenario









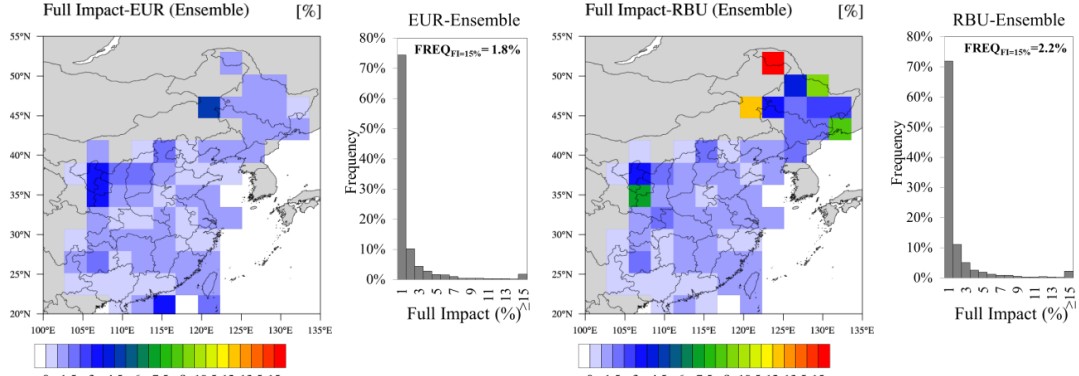

Figure 9. Spatial distributions and histograms of the long-range transport full impacts during the haze episodes.





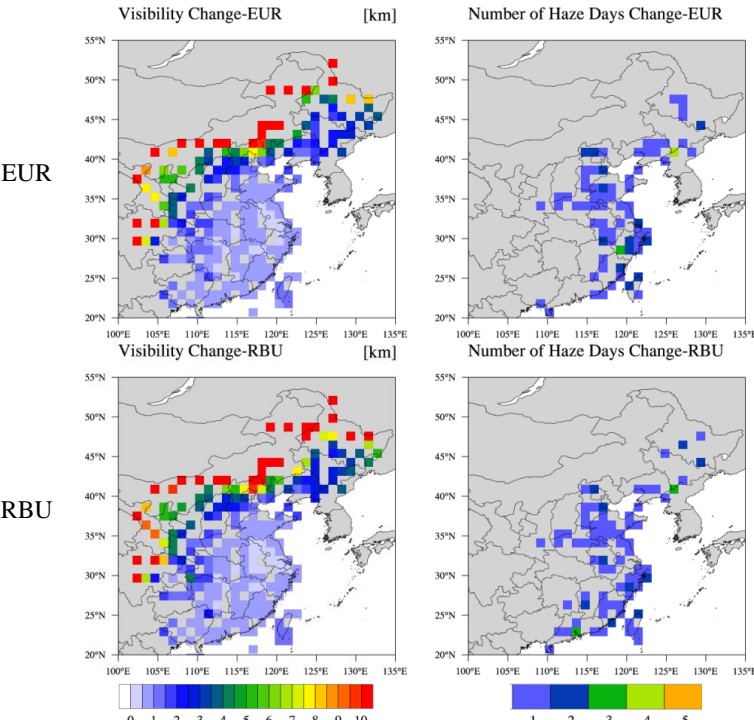

Figure 10. Reduction of visibility (left column) and enhancement of number of haze days (right column) under the EURALL (upper row) and RBUALL (lower row) scenarios.



Table 1. Models used for this study

| Model | Resolution (lat/lon/vertical) | Meteorology | Model Reference |
|---|---|---|---|
| CAM-chem | 1.9°×2.5°×56 | GEOS5 v5.2 | (Tilmes et al., 2016) |
| CHASER | 2.8°×2.8°×32 | ERA-Interim and HadISST | (Sudo et al., 2002) |
| EMEP | 0.5°×0.5°×20 | ECMWF-IFS | (Simpson et al., 2012) |
| GEOS5 | 1.0°×0.75°×72 | MERRA | Rienecker et al. (2008) |
| GEOSCHEMADJOINT | 2.0°×2.5°×72 | MERRA | (Henze et al., 2007) |
| SPRINTAS | 1.1°×1.1°×56 | ECMWF Interim | (Takemura et al., 2005) |
| Model Ensemble Mean | 2.8°×2.8°×32 | - | - |





Table 2. Annual average long-range transport impacts of surface PM concentrations and percentage contributions from the EUR and RBU source regions to the EAS receptor region. Numbers collected from the HTAP1 assessment are presented in Italic font, aerosol surface concentrations (Surf. Conc.) under the baseline scenario are presented in bold font. Numbers in the parentheses indicate the range of each variable among the participating models.

| | | Long-range transport Full Impact | | | | |
| | | EA as receptor | | EAS as receptor | | |
| | | EU→EA | EUR→EA | | | |
| | | 2000[1] | 2010EA[2] | 2008[3] | 2009[4] | 2010 |
| SO$_4^{2-}$ | **Surf. Conc. (µg/m³)** | *2.94 (1.96-4.42)* | **3.25 (2.07-5.46)** | **5.9 (5.38-6.51)** | **5.29** | **3.80 (1.45-6.67)** |
| | *Full_Impact$_{EUR\%}$* | *5.0 (0.3-9.8)* | 0.5 (0.1-0.9) | 3.5 (2.9-4.1) | 4.7 | 2.7 (0.4-5.6) |
| | *Full_Impact$_{RBU\%}$* | | | 5.5 | 5.2 | 4.1 (2.6-6.9) |
| BC | **Surf. Conc. (µg/m³)** | *0.42 (0.28-0.71)* | **0.56 (0.34-0.74)** | **1.00 (0.93-1.08)** | **0.92** | **0.82 (0.51-1.07)** |
| | *Full_Impact$_{EUR\%}$* | *1.0 (0.5-3.9)* | 0.2 (0.03-0.3) | 1.2 (0.6-1.8) | 1.9 | 1.1 (0.1-2.2) |
| | *Full_Impact$_{RBU\%}$* | | | 3.6 | 1.8 | 1.1 (0.1-2.5) |
| OA | **Surf. Conc. (µg/m³)** | *1.46 (0.81-2.52)* | **3.56 (1.93-6.29)** | **6.28 (3.51-9.06)** | **3.37** | **5.06 (2.1-8.87)** |
| | *Full_Impact$_{EUR\%}$* | *0.4 (0.2-0.9)* | 0.2 (0.02-0.4) | 0.7 (0.3-1.1) | 2.1 | 0.9 (0.1-1.2) |
| | *Full_Impact$_{RBU\%}$* | | | 2.5 | 2.0 | 1.0 (0.1-3.2) |
| | | Long-range transport Relative Impact | | | | |
| SO$_4^{2-}$+OA | *Relative_Impact$_{EUR\%}$* | 2.9 | 2.2 | 2.9 | 2.8 | 2.7 |
| | *Relative_Impact$_{RBU\%}$* | | 3.3 (2.1-5.5) | 3.8 | 3.3 | 3.7 |
| | | Local 20% anthropogenic emission perturbation impact | | | | |
| SO$_4^{2-}$+OA | $\frac{\Delta PM_{EAS}}{PM_{BASE}} \times 100\%$ | *16.8* | 12.5 | 14.0 | 14.1 | 12 |

[1] Numbers shown for 2000 are collected from the HTAP1 report that representing the long-range transport impact from EU to EA.

[2] 2010EA is calculated with the HTAP2 data by using the HTAP1 domain configuration for EA

[3] Only two models (CAM-chem and CHASER) data are available for EURALL scenario in 2008, and only one model (CAM-chem) data is available for RBUALL scenario in 2008, so no range is calculated for RBU%.

[4] Only one model (CAM-chem) 2009 data is available so no range is calculated for EUR% and RBU%.



Table 3. Long-range transport full impacts on annual average scale and during the haze episodes. Numbers in the parentheses indicate the percentage contributions.

| Models | Base PM$_{2.5}$ [µg/m$^3$] | | EUR Full Impact [µg/m$^3$ (%)] | | RBU Full Impact [µg/m$^3$ (%)] | |
|---|---|---|---|---|---|---|
| | AAVG | HAZE | AAVG | HAZE | AAVG | HAZE |
| CHASER | 20.46 | 47.73 | 0.23 (1.2) | 1.00 (2.1) | 0.29 (1.4) | 0.99 (2.1) |
| EMEP | 17.35 | 29.34 | 0.05 (0.3) | 0.11 (0.4) | 0.23 (1.3) | 0.61 (2.1) |
| GCA[1] | 25.47 | 28.03 | 0.12 (0.3) | 0.29 (1.1) | 0.35 (1.4) | 0.86 (3.0) |
| SPRINTAS | 17.45 | 24.80 | 1.00 (5.7) | 2.58 (10.5) | 1.26 (7.2) | 2.82 (11.4) |
| Ensemble | 20.18 | 32.48 | 0.35 (1.7) | 0.99 (3.1) | 0.53 (2.6) | 1.32 (4.1) |

[1]GCA: GEOSCHEMADJOINT