# Peer review of "Long-range Transport Impacts on Surface Aerosol Concentrations and the Contributions to Haze Events in China: an HTAP2 Multi-Model Study"

_Atmospheric Chemistry and Physics, 2018_

## Referee Comment (RC1) · Anonymous Referee #1 · 24 May 2018

The manuscript submitted by Dong et al. reports a basic statistical analysis of 6 simulations from HTAP2 global modelling exercise, aimed at assessing the simulated impact of long-range transport of pollutants from Europe and Russia on China's haze events. The scope of the work is well defined, I think there is some gap that may be filled in terms of link with the existing literature, and there is generally no attempt by the authors in explaining the reasons for inter-model differences. The manuscript is basically a description, sometimes lengthy, of the materials presented in the figures and the tables. Considering the relevance of the topic, I think the manuscript could be published on ACP, after considering some suggestions given below, and after careful English editing.

[Figure]

Specific comments:

1. In the introduction the authors very briefly review the literature regarding existing studies on haze in China. It is mentioned that long-range transport contribution to haze episodes is poorly documented (indeed they do not insert any reference). However, the literature on long-range transport to China is not null, and part of the phenomenology and underlying mechanisms might be in common with period of haze episodes. From a very quick literature search I identified, as potential references:

- Lee et al., Heavy metals and Pb isotopic composition of aerosols in urban and suburban areas of Hong Kong and Guangzhouâ Ă Ť Evidence of the long-range transport of air contaminants, Atmospheric Environment, Volume 41, Issue 2, January 2007, Pages 432-447

- Kong et al., Receptor modeling of PM2.5, PM10 and TSP in different seasons and long-range transport analysis at a coastal site of Tianjin, China, Science of The Total Environment, Volume 408, Issue 20, 15 September 2010, Pages 4681-4694

- Akimoto, Global Air Quality and Pollution, Science 05 Dec 2003: Vol. 302, Issue 5651, pp. 1716-1719 (and references therein)

I suggest to deepen the review of the literature on long-range transport from Europe to East Asia and put it into the fourth paragraph of the introduction. The same material might be subsequently used in the interpretation of some of the results illustrate afterwards (e.g. in section 3.1 and 3.2.

2. page 4, lines 1-5: I think these very general statements, without any specific reference, on physical processes should be avoided in the manuscript. Please add proper reference and try to be more specific on the region and the situation you are referring to.

3. section 2.2: all the data versions and source of data are missing. Please add the exact product names of the data used, the web source used, and the version of the

algorithms. This is necessary for the reproducibility of the work.

4. Figures 2 and 3 and related comments: there are some apparent inconsistency between the results presented in these figures. For example, PM2.5 is overestimated by GEOSCHEMADJOINT and underestimated by CHASER, but then AOD at AERONET sites has the opposite bias for these models. Why is that? Perhaps it could be useful to include a comparison only for some specific station for which all the datasets are available, or at least within the same model grid. From Figure 1 it seems to be possible for some stations.

5. Figure 4 and related comments: the modelled AOD over China and elsewhere in the domain differ among models by more than a factor of two. As for previous results on point measurements, there is no attempt to explain the differences. For example, considering the same anthropogenic emissions, the difference over China CHASER and SPRINTARS is quite remarkable.

6. Figures 5-6 and related comments. The figures are interesting because they nicely illustrate the model diversity. For example, the seasonal cycle of contributions from some models is opposite to that of others (e.g. CAM-Chem peaks in summer, CHASER in winter, and GEOS5 in spring). It would be useful to have some inspection of these difference. I suspect that differences in the meteorological fields used in these models are responsible for the large variability.

7. Figures 9-10: some panels look patchy, for example EMEP, SPRINTARS and all in Figure 10. Why is that?

8. I recommend English editing of the manuscript. The use of language is imaginative and makes understanding difficult. A few random examples:

- p. 3, l. 40-41: "These datasets are essential to estimate surface PM response compare the aerosol transport in different atmosphere layers". What is "response compare"? "atmosphere" –> "atmospheric"

- p. 5. l. 6-7: "the models all tend to underestimate the high peaks in spring (Mar.-Apr.) and low bottoms in summer". Not clear what "low bottoms" means.

- note 2 on caption of Figure 2: "PM2.5 observations in EUR and EAS region have no standard because there are no sites with valid measurements fall into the same model ensemble mean grid". Very difficult to understand: why a standard deviation cannot be calculated even if stations are not in the same model cell?

---

## Referee Comment (RC2) · Anonymous Referee #2 · 3 Jul 2018

This paper presented a work analyzing the contribution from Europe to China's atmospheric particle concentrations and haze events, with intensive chemistry transport modeling. The authors made great efforts on incorporating multiple transport models to understand the difference between models and to reduce the uncertainty of simulation. They also evaluated the impacts of emission inventory on the simulation, as the accuracy of emission inventory for anthropogenic pollutants is always a big concern on the air quality research community. Before it can be accepted as a final atmospheric chemistry physics paper, however, the following issues need to be further discussed or stressed.

[Figure]

1. The significance of the paper needs to be reconsidered and relevant statement should be revised. In current format, the authors stated that there were limited studies conducted on regional transport to China and it might be important as China is controlling its emissions. The results, however, show that the impacts of Europe was very few, and the studying period was before 2010, during which the emissions in East Asia were expected to still increase. It seems that the current work did not fully answer the question they raisedïij§The most serious haze events after 2010 were not included in this study?

2. Lines 9-10, Page 2: this results is quite old, there are recently more studies on health impacts of China's air pollution.

3. Lines 13-16, Page 6: Please define how the MNB was calculated. Is there any criterion indicating the acceptable range of MNB?

4. There are limited PM2.5 observations used in model evaluation for China. I understand that the official data were not available until 2013. However, could the data published in previous studies be available and could the evaluation be improved?

5. Figures 5 and 6 illustrated the surface aerosol response under EURALL and RBUALL. Can you explain why the seasonal variations were different between models? In Fig 6, for example, larger response was found in summer for CAM-chem and SPRINTARS, while smaller was found in summer in EMEP and GEOSCHEMADJOINT. By the way, caption of Figure 6 should be revised (Figure XXX?).

6. Lines 23-27, Page 10. I am not persuaded by the authors by the linearity assumption. They estimated the full impact by scaling the PM responses under the 20% emissions perturbation conditions by a factor of 5. If this is the case, why not directly estimate the PM response by removing 100% emissions for given region? Nonlinearity of PM2.5 concentrations to precursor emissions was strong, and the uncertainty of the assumption should be carefully analyzed and quantified.

---

## Referee Comment (RC3) · Anonymous Referee #3 · 6 Jul 2018

The manuscript submitted by Dong et al. assesses how changes in aerosol emissions in Europe and Russia influences haze events in China, using simulations from the HTAP2 project. Analyses include a thorough model evaluation towards various surface- and satellite-based observation data, presentations of the seasonality of the long-range impacts from the two regions on China, evaluation of how the long-range impacts are distributed between within- and above-PBL layers, comparison of results to findings for earlier years, as well as an analysis of estimated horizontal visibility and how this variable is affected by the two source regions. The authors have performed many and rigorous analyses, and the results are likely to be of broad interest to the community. There are, however, some issues that need to be resolved before the paper should be

accepted for publishing.

General comments: -The language of the manuscript could greatly benefit from a thorough read-through by a person fluent in English. - The manuscript is at times unnecessarily lengthy. I have suggested several sentences that could be removed, but going through the manuscript and removing sentences and statements that contain irrelevant information or information that has already been given, will help the reader. - In the Introduction, it would be good to see a bit more background on haze in China – for instance, write out in more detail what the references around lines 15-20 find. Do that Wang studies referred to on line 19 look at sources in China only, or is there an element of long-range influences here that could be relevant for this study? - The "Results and Discussion" section is at times too much description of figures and numbers, and too little discussion of results. I believe a lot of the numbers could be put in a table so that more time can be spent on the main highlights and how they agree/differ from other findings. There are several interesting results and features here that deserve to be accentuated.

Specific comments: - P1 L37: add "from EUR" after "long-range transport"? - P1 L38: change "aerosol response" to "the aerosol response in EAS" - P1 L44: to compare how much 1-3 days change in haze frequence is to the percentages given above, please consider changing 1-3 days to percent change - P2 L12: It is a bit difficult to catch the meaning of the sentence starting with "Although" – a rewording would be good! - P2 L28: Not necessary to introduce the AQMEII and MICS-Asia projects, as data form these are not used in the present study? Instead, line 26 could instead start with "One of these is the Task Force on Hemspheric..." - P2 L34: These last two sentences are not strictly necessary. - P3 L27: The first part of this sentence "To quantify …...sensitivity simulations, " is superfluous – one could instead start directly at "Emission perturbations are conducted with all.." - P3 L31: Fix reference Guido R. can der Werf? - P3 L41: the sentence starting with "These datasets are essential" can also be removed. - P4 L2: Here you could stop after "descend into the PBL." and

then start a new sentence motivating the remaining text by stating the relevance of the PBL-analysis to haze (for instance, that pollutants within the PBL give more haze, and therefore it is necessary to understand the contributions of within- and above-PBL) - P4 L37: I may have missed something, but P3 L19 says that all models have resolution of 0.1x0.1 – where does the 2.8x2.8 come from? Please clarify. - P5 L1: Please define MB - P5 L5: Please consider replacing all uses of "temporal" in this section with "seasonal", as the "temporal" gives an impression of temporal (year-to-year) development. - P5 L6: you write that models tend to underestimate the high peaks in spring, but Fig. 2d seems to me to show that models _over_estimate in spring (or at least all models are higher in March, and the observations are in the midst of the models in April)? - P5 L15: Remove "shows significantly . . . than the others", which is given from the previous sentence. - P5 L16: Do you have any data on the occurrence or tendency for wildfires near this specific stations? If not, this comment should perhaps be removed. - P9 Section 3.3 heading: I am a bit skeptical to the use of the word "Trend" in this heading and in the section text, as a trend can hardly be quantified based on a comparison between the years 2000 and 2010 (data for years 2008 and 2009 helps, but the data are scarce). Consider changing "trend" to "change" or something similar. - P9 L39: Please add a reference after "the past decade." - P11 L25: How would the results look if you use CAM-chem only for all the years? - P12 L16: ECE –> CEC? - P13 L10: "The participating models. . . to 5.5%" can be removed as it has just been said above. - P13 L14: It says Frequency_Full_Impact15 twice :) - P13 L34: Please give this in % change as well. - P14 L37: Please add references after "recent years".
* * *

---

## Author Comment (AC1) · 17 Aug 2018

**General comment**: The manuscript submitted by Dong et al. reports a basic statistical analysis of 6 simulations from HTAP2 global modelling exercise, aimed at assessing the simulated impact of long-range transport of pollutants from Europe and Russia on China's haze events. The scope of the work is well defined, I think there is some gap that may be filled in terms of link with the existing literature, and there is generally no attempt by the authors in explaining the reasons for inter-model differences. The manuscript is basically a description, sometimes lengthy, of the materials presented in the figures and the tables. Considering the relevance of the topic, I think the manuscript could be published on ACP, after considering some suggestions given below, and after careful English editing.
**Response**: We appreciate the referee for the overall positive comment and providing the helpful detailed suggestions. We have rewrote unnecessary long sentences and read through the whole manuscript for English editing.

Specific comments:
1. In the introduction the authors very briefly review the literature regarding existing studies on haze in China. It is mentioned that long-range transport contribution to haze episodes is poorly documented (indeed they do not insert any reference). However, the literature on long-range transport to China is not null, and part of the phenomenology and underlying mechanisms might be in common with period of haze episodes. From a very quick literature search I identified, as potential references:
- Lee et al., Heavy metals and Pb isotopic composition of aerosols in urban and suburban areas of Hong Kong and Guangzhou, South China. Evidence of the long-range transport of air contaminants, Atmospheric Environment, Volume 41, Issue 2, January 2007, Pages 432-447
- Kong et al., Receptor modeling of $PM_{2.5}$, $PM_{10}$ and TSP in different seasons and long-range transport analysis at a coastal site of Tianjin, China, Science of The Total Environment, Volume 408, Issue 20, 15 September 2010, Pages 4681-4694
- Akimoto, Global Air Quality and Pollution, Science 05 Dec 2003: Vol. 302, Issue 5651, pp. 1716-1719 (and references therein)
I suggest to deepen the review of the literature on long-range transport from Europe to East Asia and put it into the fourth paragraph of the introduction. The same material might be subsequently used in the interpretation of some of the results illustrate afterwards (e.g. in section 3.1 and 3.2).
**Response**: The references listed above are closely related to our study thus they have been properly cited in the revised manuscript. The objective of this study is to evaluate the contribution of long-range transport to $PM_{2.5}$, with special focus on the haze episode. Previous studies about long-range transport mainly focused on the exported air pollutants from China to other areas or the transport of O3. Although some studies (e.g., the Akimoto 2003 publication, and the HTAP Phase1 report) pointed out that mitigating global air pollution requires international participations of multiple countries or continents, the contribution of long-range transport to $PM_{2.5}$ in China remains poorly documented. The references suggested by the referee are very helpful. We also added a more detailed description of the research status about long-range transport of air pollutants to China.

2. page 4, lines 1-5: I think these very general statements, without any specific reference, on physical processes should be avoided in the manuscript. Please add proper reference and try to be more specific on the region and the situation you are referring to.
**Response**: Thanks for the reminder, we have added proper references (Eckhardt et al., 2003; Sthol et al., 2002) and related descriptions.

3. section 2.2: all the data versions and source of data are missing. Please add the exact product names of the data used, the web source used, and the version of the algorithms. This is necessary for the reproducibility of the work.

**Response**: Thanks for the comment, we have added all the products names, versions, and web sources in the revised manuscript. These details are also summarized in the following table (API, EANET, and EBAS has no version updates information, the data is downloaded from the web source).

Table. Version details and web sources of the data used for model evaluation

| Data used | Web source |
|---|---|
| AERONET (Level2.0, version2) | http://aeronet.gsfc.nasa.gov |
| API | http://datacenter.mep.gov.cn |
| EANET | http://www.eanet.asia/ |
| EBAS | http://ebas.nilu.no |
| MODIS (MOD08, MYD08) | https://modis.gsfc.nasa.gov |

4. Figures 2 and 3 and related comments: there are some apparent inconsistency between the results presented in these figures. For example, $PM_{2.5}$ is overestimated by GEOSCHEMADJOINT and underestimated by CHASER, but then AOD at AERONET sites has the opposite bias for these models. Why is that? Perhaps it could be useful to include a comparison only for some specific station for which all the datasets are available, or at least within the same model grid. From Figure 1 it seems to be possible for some stations.

**Response**: This is a very interesting point and we thank the referee for mention it. Figure 2 shows PM2.5 was overestimated by GEOSCHEMADJOINT by 7.5 µg/m³ (63%) in EAS (EANET and API stations), by 8.6 µg/m³ (66%) in EUR (EBAS stations). Figure 3 suggests that GEOSCHEMADJOINT underestimates AERONET-AOD by -0.08 (-23%) in EAS and overestimates AOD by 0.004(4%) in EUR. As suggested by the referee, we selected EANET-Oki (36.28ºN, 133.18ºE) as the PM2.5 site and AERONET-Osaka (34.65ºN, 135.59ºE) as the AOD site in EAS region, and selected EBAS-Revin (49.90ºN, 4.63ºE) as the PM2.5 site and AERONET-Brussels (50.78ºN, 4.35ºE) as the AOD site in EUR region. These are the closest nearby sites in each of the domain. Simulation bias of GEOSCHEMADJOINT at these sites are shown in the following figure.

[Figure]

Figure. Simulation bias of GEOSCHEMADJOINT for $PM_{2.5}$ (solid red circles) and AOD (solid blue squares)

As shown in the figure, GEOSCHEMADJOINT overestimated $PM_{2.5}$ but underestimate AOD throughout the full year 2010 in EAS region. We examined this issue and found out there are two reasons: first, there are relatively less $PM_{2.5}$ observation sites (2 in EAS, 5 in EUR) compared to large number of AOD observation sites (15 in EAS, 73 in EUR). The EANET-Oki station was located on a small island ~50 miles from west coast of Japan thus represents the background concentration, while the AERONET-Osaka site is located in the downtown area of Osaka City. Evaluation in EUR region has the similar condition, the EBAS-Revin site is in a national park, while the AERONET-Brussels site is close to downtown. Although some AERONET sites are also located in remote areas, it generally has a more comprehensive representation of different surroundings including both rural and urban, but the $PM_{2.5}$ data

used in this study are most located in rural area. Second, GEOSCHEMADJOINT are reported as tend to overestimate the surface layer $PM_{2.5}$ concentration in Asia (Figure 2 in Gu and Liao, 2016; Figure 2 in Xu et al., 2015) and underestimate the column density AOD (Figure 4 in Choi et al,. 2009) in East Asia, although the explicit reason for this inconsistency hasn't been well documented. So generally the performance and evaluation results of this HTAP Phase 2 modeling effort is consistent with those previous studies.

References:

Yi-Xuan GU & Hong LIAO (2016) Response of fine particulate matter to reductions in anthropogenic emissions in Beijing during the 2014 Asia–Pacific Economic Cooperation summit, Atmospheric and Oceanic Science Letters, 9:6, 411-419, DOI: 10.1080/16742834.2016.1230465

Xu, J.-W., Martin, R. V., van Donkelaar, A., Kim, J., Choi, M., Zhang, Q., Geng, G., Liu, Y., Ma, Z., Huang, L., Wang, Y., Chen, H., Che, H., Lin, P., and Lin, N.: Estimating ground-level PM2.5 in eastern China using aerosol optical depth determined from the GOCI satellite instrument, Atmos. Chem. Phys., 15, 13133-13144, https://doi.org/10.5194/acp-15-13133-2015, 2015.

5. Figure 4 and related comments: the modelled AOD over China and elsewhere in the domain differ among models by more than a factor of two. As for previous results on point measurements, there is no attempt to explain the differences. For example, considering the same anthropogenic emissions, the difference over China CHASER and SPRINTARS is quite remarkable.

**Response**: We also notice the large difference between model performances with the same emission inputs. Explicitly clarify the causes of the difference would require deep detailed investigation of the model schemes, algorithms, and parameterization, which is not within the scope of this study. But the other HTAP Phase2 related studies (Im et al., 2018; Palacios-Peña et al., 2018; Astitha et al., 2018) do present a few investigations into the multi-model comparison between the models used in this study, and the different model performance are attributed to meteorology (in particular wind speed and PBL height), different aerosol mechanisms, treatment of wind-blown dust emission, and biomass burning emission injection heights. Previous multi-modeling efforts such as the AEROCOM also pointed out that these aspects can lead to modeled AOD and surface PM concentration differ by a factor of 2 and 10 respectively, although the some AEROCOM participating models are different from HTAP. We agree with the referee that briefly explain the difference is necessary as our discussion is based on multi-model simulations, so we have added a short discussion in the revised manuscript. The above-mentioned references are also added into the revised manuscript.

References:

Im, U., Christensen, J. H., Geels, C., Hansen, K. M., Brandt, J., Solazzo, E., Alyuz, U., Balzarini, A., Baro, R., Bellasio, R., Bianconi, R., Bieser, J., Colette, A., Curci, G., Farrow, A., Flemming, J., Fraser, A., Jimenez-Guerrero, P., Kitwiroon, N., Liu, P., Nopmongcol, U., Palacios-Peña, L., Pirovano, G., Pozzoli, L., Prank, M., Rose, R., Sokhi, R., Tuccella, P., Unal, A., Vivanco, M. G., Yarwood, G., Hogrefe, C., and Galmarini, S.: Influence of anthropogenic emissions and boundary conditions on multi-model simulations of major air pollutants over Europe and North America in the framework of AQMEII3, Atmos. Chem. Phys., 18, 8929-8952, https://doi.org/10.5194/acp-18-8929-2018, 2018.

Palacios-Peña, L., Jiménez-Guerrero, P., Baró, R., Balzarini, A., Bianconi, R., Curci, G., Landi, T. C., Pirovano, G., Prank, M., Riccio, A., Tuccella, P., and Galmarini, S.: Aerosol optical properties over Europe: an evaluation of the AQMEII Phase 3 simulations against satellite observations, Atmos. Chem. Phys. Discuss., https://doi.org/10.5194/acp-2017-1119, in review, 2018.

Astitha, M., Kioutsoukis, I., Fisseha, G. A., Bianconi, R., Bieser, J., Christensen, J. H., Cooper, O., Galmarini, S., Hogrefe, C., Im, U., Johnson, B., Liu, P., Nopmongcol, U., Petropavlovskikh, I., Solazzo, E., Tarasick, D. W., and Yarwood, G.: Seasonal ozone vertical profiles over North America using the AQMEII group of air quality models: model inter-comparison and stratospheric intrusions, Atmos. Chem. Phys. Discuss., https://doi.org/10.5194/acp-2018-98, in review, 2018.

6. Figures 5-6 and related comments. The figures are interesting because they nicely illustrate the model diversity. For example, the seasonal cycle of contributions from some models is opposite to that of others (e.g. CAM-Chem peaks in summer, CHASER in winter, and GEOS5 in spring). It would be useful to have some inspection of these difference. I suspect that differences in the meteorological fields used in these models are responsible for the large variability.

**Response**: CAM-chem showed the largest PM response in summer under EUR emission reduction scenario, and SPRINTARS showed the largest PM response in summer under RBU emission reduction scenario, while the other models all showed larger PM responses in winter or spring. We agree with the referee that meteorology difference might be one of the reasons for simulation diversity. We examined the surface air temperature used by the participating models. Domain averages of monthly temperature over EUR and RBU are shown in the following figure.

[Figure]

Figure. Monthly surface air temperature from CAM-chem, CHASER, and GEOS5

In EUR region, CAM-chem and SPRINTARS simulated surface air temperatures are systematically higher than other models by ~2.5K in winter. A higher temperature in the emission source region may facilitate the PM precursors' chemistry and subsequently allow less precursors enter long-range transport. In RBU region, SPRINTARS simulated temperature is ~2K lower than other models in summer, which may lead to more precursors transport into EAS and subsequently induce larger PM response. But on the other hand, temperature is apparently not the only influencing factor as CAM-chem showed highest temperature in summer over EUR region yet largest PM response in summer too. Wind speed and PBL height may play a more important role as indicated by Im et al. (2018), but unfortunately only one of the participating model provided wind and PBL data. Explicitly identify the impact of meteorology on modeled PM response would require a set of more detailed experiments, and this is beyond the scope of HTAP program. CAM-chem applied the modified Zhang-McFarlane approach (Zhang and McFarlane, 1995) with shallow convection follows Hack et al. (2006). GEOS5 applied the modified scheme by Moorthi and Suarez (1992), which is a relaxed Arakawa-Schubert algorithm. These schemes are functionalized and parameterized substantially different and will subsequently lead to differences of aerosol vertical distribution, lifetime, transport, and total suspended aerosol concentration in the atmosphere (Stjern et al., 2016). Aerosol parameterization also lead to different $PM_{2.5}$ formula. CAM-chem simulates secondary organic aerosol (SOA) with the 2-product approach using laboratory-determined yields from photooxidation of monoterpenes, isoprene and aromatics, while GEOS5 has no SOA. The differences of aerosol scheme, heterogeneous chemistry, treatment of OC, OA, and SOA lead to additional inter-model variability. In addition, grid resolutions diversity is also responsible as Molod et al (2015) demonstrated that different grid resolutions will result in different scavenging aerosol even with the same model. In fact, not only the $PM_{2.5}$ responses but also the baseline $PM_{2.5}$ concentrations show prominent different seasonality among the models in both the HTAP Phase1 (Dentener et al., 2010) and Phase2 program, and this is also why multi-model mean is used to estimate the source-receptor relationship. We have added the abovementioned discussion and references in the revised manuscript.

7. Figures 9-10: some panels look patchy, for example EMEP, SPRINTARS and all in

Figure 10. Why is that?

**Response**: Figures 9-10 are designed to demonstrate the full impact of long-range transport during the haze episode, so the NCDC surface observation data is used to identify where and when haze exist. For those with finer grid resolution such as EMEP (0.5×0.5º) and SPRINTARS (1.1×1.1º), there are some model grids having no NCDC observation site, and these grids are filled with missing value, and this makes the figures look patchy. Although haze (visibility) can also be estimated with aerosol extinction coefficient, using the direct measurements from NCDC is apparently a more solid method to identify haze, and only SPRINTARS provides the aerosol extinction coefficient data. We have added a short sentence in the figure caption to explain the patchy panels to avoid misleading.

8. I recommend English editing of the manuscript. The use of language is imaginative and makes understanding difficult. A few random examples:

- p. 3, l. 40-41: "These datasets are essential to estimate surface PM response compare the aerosol transport in different atmosphere layers". What is "response compare"? "atmosphere" –> "atmospheric"

**Response**: We agree that English editing is necessary, it's also pointed out by another referee. This sentence is removed because it is not necessary.

- p. 5. l. 6-7: "the models all tend to underestimate the high peaks in spring (Mar.-Apr.) and low bottoms in summer". Not clear what "low bottoms" means.

**Response**: The sentence is changed to "Temporal variation of $O_3$ is also simulated well in EAS, although the models all tend to underestimate the high values in spring (Mar.-Apr.) and low concentrations in summer (Jul.-Sep.)"

- note 2 on caption of Figure 2: "PM2.5 observations in EUR and EAS region have no standard because there are no sites with valid measurements fall into the same model ensemble mean grid". Very difficult to understand: why a standard deviation cannot be calculated even if stations are not in the same model cell?

**Response**: The standard deviation is calculated between the observations from different sites in the same model grid, we have mentioned in the caption of Figure 2 that "vertical error bars depict the standard deviation across the sites in the same ensemble grid."

---

## Author Comment (AC2) · 17 Aug 2018

**General comment**: This paper presented a work analyzing the contribution from Europe to China's atmospheric particle concentrations and haze events, with intensive chemistry transport modeling. The authors made great efforts on incorporating multiple transport models to understand the difference between models and to reduce the uncertainty of simulation. They also evaluated the impacts of emission inventory on the simulation, as the accuracy of emission inventory for anthropogenic pollutants is always a big concern on the air quality research community. Before it can be accepted as a final atmospheric chemistry physics paper, however, the following issues need to be further discussed or stressed.
**Response**: We thank the referee for the encouraging comment and providing insightful suggestions.

1. The significance of the paper needs to be reconsidered and relevant statement should be revised. In current format, the authors stated that there were limited studies conducted on regional transport to China and it might be important as China is controlling its emissions. The results, however, show that the impacts of Europe was very few, and the studying period was before 2010, during which the emissions in East Asia were expected to still increase. It seems that the current work did not fully answer the question they raised. The most serious haze events after 2010 were not included in this study?
**Response**: We have added some references documenting the long-range transport into China (Lee et al., 2007; Kong et al., 2010; Akimoto 2003; Fu et al., 2012) and a short introduction of their findings. In our study, Table 2 summarized the annual average long-range transport contribution from EUR and RBU regions to EAS region in year 2000 and 2008-2010. Table 3 summarized the long-range transport contributions during the haze episode. We raised the research question in "Introduction" section that "the background concentrations of PM and the contributions from outside China import of air pollutants to the haze problem, is poorly documented." So the question is answered by Tables 2-3 and the related discussions. The severe haze event in 2010 is included in this study but not specifically highlighted. Some places in China has more than 300 days with haze identified with NCDC observation. We analyzed the annual total haze events and reported the contribution of long-range transport to these events, as shown with Figures 9-10. An overview of the haze events for full year 2010 is provided in supplementary material Figure S1.

2. Lines 9-10, Page 2: this results is quite old, there are recently more studies on health impacts of China's air pollution.
**Response**: We have added several most recent studies that reported the premature deaths attributable to $PM_{2.5}$ pollution in China from 2013 to 2017, these references include: Huang et al., 2018; Cao et al., 2017; Li et al., 2018.

3. Lines 13-16, Page 6: Please define how the MNB was calculated. Is there any criterion indicating the acceptable range of MNB?
**Response**: The MNB is calculated as:

$$MNB = \frac{1}{S}\sum_{i=1}^{S}\sum_{j=1}^{T}\frac{sim_{ij} - obs_{ij}}{obs_{ij}}$$

Where $S$ is the number of observation stations, $T$ is the total number of month, $obs_{ij}$ is the observed value at the station $i$ and month $j$, and $sim_{ij}$ is the corresponding simulation value at the closet grid point to the station. There is no well documented threshold for an acceptable MNB, but the AERCOM research program have been frequently cited by the research community, so we used their MNB values to demonstrate our participating models' performance.

4. There are limited PM$_{2.5}$ observations used in model evaluation for China. I understand that the official data were not available until 2013. However, could the data published in previous studies be available and could the evaluation be improved?

**Response**: This is a very insightful comment and we agree that there many some publications reporting the measured PM$_{2.5}$ concentrations in China (Zhang and Cao, 2016; Lowsen and Conway, 2016), but the diversities in instrument, measuring method, and sampling period make it difficult to develop a consistent observation database from the literatures. In addition to the potential uncertainties within each individual measurement literature, these measurements are usually presented with charts or figures so we would have to roughly read the values from the figures, which may introduce more uncertainty and is not proper for the HTAP program as it requires applying official downloadable data by all participating groups so all experiments and analysis could be reproduced. In contrast, the EANET data used in this study provides measurement collected with the same type of instrument and method. Considering the limited number of EANET sites, we also included AERONET and MODIS AOD which are all public accessible for model evaluation with better spatial and temporal coverage. In addition, since examining long-range transport of surface PM$_{2.5}$ into China is the main objective of this study, evaluating models performances with literature review collected data would require intensive efforts and make the manuscript lengthy.

5. Figures 5 and 6 illustrated the surface aerosol response under EURALL and RBUALL. Can you explain why the seasonal variations were different between models? In Fig 6, for example, larger response was found in summer for CAM-chem and SPRINTARS, while smaller was found in summer in EMEP and GEOSCHEMADJOINT. By the way, caption of Figure 6 should be revised (Figure XXX?).

**Response**: We apologize for the typo in the figure caption, it is corrected in the revised manuscript. We agree with the referee that prominent different seasonality was found between CAM-chem and other models. Despite applying the same emission inputs, several aspects of the participating models lead to the different seasonality of PM2.5 response. One of the other two referees also pointed out this issue, and we briefly probe into these aspects. These aspects including the meteorology, aerosol mechanisms, and convection mechanisms. We first examined the meteorology differences by comparing the model simulated air temperature the following figure shows domain average monthly mean surface air temperature from CAM-chem, CHASER, GEOS5, and SPRINTARS. In EUR region, CAM-chem and SPRINTARS simulated surface air temperatures are systematically higher than other models by ~2.5K in winter. A higher temperature in the emission source region may facilitate the PM precursors' chemistry and subsequently allow less precursors enter long-range transport. In RBU region, SPRINTARS simulated temperature is ~2K lower than other models in summer, which may lead to more precursors transport into EAS and subsequently induce larger PM response. But on the other hand, temperature is apparently not the only influencing factor as CAM-chem showed highest temperature in summer over EUR region yet largest PM response in summer too. Wind speed and PBL height may play a more important role as indicated by Im et al. (2018) but unfortunately only one of the participating model provided wind and PBL data.

[Figure]

[Figure]

Figure. Monthly surface air temperature from CAM-chem, CHASER, and GEOS5

We then examined the convection schemes among models. CAM-chem applied the modified Zhang-McFarlane approach (Zhang and McFarlane, 1995) with shallow convection follows Hack et al. (2006). GEOS5 applied the modified scheme by Moorthi and Suarez (1992), which is a relaxed Arakawa-Schubert algorithm. These schemes are functionalized and parameterized substantially different and will subsequently lead to differences of aerosol vertical distribution, lifetime, transport, and total suspended aerosol concentration in the atmosphere (Stjern et al., 2016). Aerosol parameterization also lead to different PM2.5 formula. CAM-chem simulates secondary organic aerosol (SOA) with the 2-product approach using laboratory-determined yields from photooxidation of monoterpenes, isoprene and aromatics, while GEOS5 has no SOA. The differences of aerosol scheme, heterogeneous chemistry, treatment of OC, OA, and SOA lead to additional inter-model variability. In addition, grid resolutions diversity is also responsible as Molod et al (2015) demonstrated that different grid resolutions will result in different scavenging aerosol even with the same model. In short summary, the abovementioned aspects may all contribute to the different seasonality of PM2.5 response, and more a set of more specifically designed model experiments is necessary to explicitly identify their influences, yet this is beyond the current scope of HTAP program. We have added a short discussion of the seasonality in the revised manuscript to point out this issue with the clues mentioned here.

References:
Hack, J. J., Caron, J. M., Yeager, S. G., Oleson, K. W., Holland, M. M., Truesdale, J. E., and Rasch, P. J.: Simulation of the Global Hydrological Cycle in the CCSM Community Atmosphere Model Version 3 (CAM3): Mean Features, J. Climate, 19, 2199–2221, doi:10.1175/JCLI3755.1, 2006.
Molod, A., Takacs, L., Suarez, M., and Bacmeister, J.: Development of the GEOS-5 atmospheric general circulation model: evolution from MERRA to MERRA2, Geosci. Model Dev., 8, 1339–1356, doi:10.5194/gmd-8-1339-2015, 2015.
Moorthi, S. and Suarez, M. J.: Relaxed Arakawa-Schubert. A Parameterization of Moist Convection for General Circulation Models, Mon. Weather Rev., 120, 978–1002, doi:10.1175/1520-0493(1992)120<0978:RASAPO>2.0.CO;2, 1992.
Zhang, G. J. and McFarlane, N. A.: Sensitivity of climate simulations to the parameterization of cumulus convection in the Canadian climate center general circulation model, Atmos.-Ocean, 33, 407–446, 1995.

6. Lines 23-27, Page 10. I am not persuaded by the authors by the linearity assumption. They estimated the full impact by scaling the PM responses under the 20% emissions perturbation conditions by a factor of 5. If this is the case, why not directly estimate the PM response by removing 100% emissions for given region? Nonlinearity of $PM_{2.5}$ concentrations to precursor emissions was strong, and the uncertainty of the assumption should be carefully analyzed and quantified.

**Response**: We agree with the referee that $PM_{2.5}$ concentrations to precursor emissions are strong, and this is the merit of applying atmospheric models to simulate the "real" aerosol response instead of simply estimating it with a certain emission change ratio. The 20% emission perturbation is the first priority of model experiment designed by the HTAP Phase2 program because it is a relatively reasonable and applicable control rate for air quality management. The impact of long-range transport however, indicates the overall contribution of the total emission in the source regions, so 100% emission reduction would be a stronger but unrealistic experiment. While the 100% emission perturbation simulation is not available, the "full impact" calculated from 20% emission perturbation is the best estimates we can derive. This method was applied by several investigations for estimating inter-continental transport of $O_3$ (Fiore et al., 2009; Lin et al., 2010; West et al., 2009; Zhang et al., 2009), which has an even more significant nonlinear response to the precursors. But as the referee mentioned, this method may introduce uncertainty due to the nonlinear response, and we also noticed this issue while analyzing the modeling data. We have applied the Response Surface Method (RSM) with Hemispheric-CMAQ model to quantify the sourcereceptor relationship with more detailed simulation design than HTAP. We are analyzing the data now and expect to report the findings later.

---

## Author Comment (AC3) · 17 Aug 2018

**General comments**: The manuscript submitted by Dong et al. assesses how changes in aerosol emissions in Europe and Russia influences haze events in China, using simulations from the HTAP2 project. Analyses include a thorough model evaluation towards various surface and satellite-based observation data, presentations of the seasonality of the long-range impacts from the two regions on China, evaluation of how the long-range impacts are distributed between within- and above-PBL layers, comparison of results to findings for earlier years, as well as an analysis of estimated horizontal visibility and how this variable is affected by the two source regions. The authors have performed many and rigorous analyses, and the results are likely to be of broad interest to the community. There are, however, some issues that need to be resolved before the paper should be accepted for publishing.

**General comments**: -The language of the manuscript could greatly benefit from a thorough read-through by a person fluent in English. - The manuscript is at times unnecessarily lengthy. I have suggested several sentences that could be removed, but going through the manuscript and removing sentences and statements that contain irrelevant information or information that has already been given, will help the reader.

- In the Introduction, it would be good to see a bit more background on haze in China – for instance, write out in more detail what the references around lines 15-20 find. Do that Wang studies referred to on line 19 look at sources in China only, or is there an element of long-range influences here that could be relevant for this study?

**Response**: We intended to provide a brief introduction of the research topics related with haze in China, so detailed findings of these references are not described. The findings of these publications are not used in our study. We listed them in the "Introduction" section as part of the background information. None of the literatures intended to quantify the contribution of long-range transport to haze in China, and this is why we conduct our study.

- The "Results and Discussion" section is at times too much description of figures and numbers, and too little discussion of results. I believe a lot of the numbers could be put in a table so that more time can be spent on the main highlights and how they agree/differ from other findings. There are several interesting results and features here that deserve to be accentuated.

**Response**: We agree with the referee that language of the original submission shall be greatly improved, and thank the referee for providing other detailed suggestions and comments. We have carefully go through the whole manuscript to improve the writing, by removing some of the descriptions of the numbers and adding more in-depth discussions, such as the possible reason for model diversity.

Specific comments:

1. P1 L37: add "from EUR" after "long-range transport"?
**Response**: It has been added in the revised manuscript.

2. P1 L38: change "aerosol response" to "the aerosol response in EAS"
**Response**: It has been added in the revised manuscript.

3. P1 L44: to compare how much 1-3 days change in haze frequency is to the percentages given above, please consider changing 1-3 days to percent change
**Response**: The percent change has been added in the revised manuscript.

4. P2 L12: It is a bit difficult to catch the meaning of the sentence starting with "Although" – a rewording would be good!

**Response**: The long sentence has been revised as "Some pilot studies have tried to explore the understanding of haze in China."

5. P2 L28: Not necessary to introduce the AQMEII and MICS-Asia projects, as data form these are not used in the present study? Instead, line 26 could instead start with "One of these is the Task Force on Hemispheric …"
**Response**: The description of AQMEII and MICS-Asia is removed and the sentence has been revised.

6. P2 L34: These last two sentences are not strictly necessary.
**Response**: These sentences are removed.

7. P3 L27: The first part of this sentence "To quantify … sensitivity simulations" is superfluous – one could instead start directly at "Emission perturbations are conducted with all.."
**Response**: That sentence is changed according to the comment.

8. P3 L31: Fix reference Guido R. can der Werf?
**Response**: The reference is removed, it shall be: "(1) BASE scenario with all baseline emissions"

9. P3 L41: the sentence starting with "These datasets are essential" can also be removed.
**Response**: The sentence is removed.

10. P4 L2: Here you could stop after "descend into the PBL." And then start a new sentence motivating the remaining text by stating the relevance of the PBL-analysis to haze (for instance, that pollutants within the PBL give more haze, and therefore it is necessary to understand the contributions of within- and above- PBL).
**Response**: These two sentences have been revised according to the comment, and necessary references are added according to the suggestion of another referee.

11. P4 L37: I may have missed something, but P3 L19 says that all models have resolution of 0.1x0.1 – where does the 2.8x2.8 come from? Please clarify.
**Response**: Pag4-Line19 states the resolution of the emission inventory, to avoid misleading, that sentence is changed to "The emissions are compiled from several regional inventories for the year 2010 with monthly temporal resolution and $0.1° \times 0.1°$ grid resolution". Page4-Line37 states that the grid resolution of ensemble mean is $2.8° \times 2.8°$, that sentence is changed to " … model ensemble mean, calculated as the average of all participating models at $2.8° \times 2.8°$ grid resolution."

12. P5 L1: Please define MB
**Response**: "MB" refers to "mean bias", it has been added in the updated version.

13. P5 L5: Please consider replacing all uses of "temporal" in this section with "seasonal", as the "temporal" gives an impression of temporal (year-to-year) development.
**Response**: It has been replaced in the updated version.

14. P5 L6: you write that models tend to underestimate the high peaks in spring, but Fig. 2d seems to me to show that models overestimate in spring (or at least all models are higher in March, and the observations are in the midst of the models in April)?
**Response**: The referee is correct, we intended to emphasize that "model overestimate high values in spring", it has been fixed in the updated version.

15. P5 L15: Remove "shows significantly … than the others", which is given from the previous sentence.
**Response**: The sentence has been removed.

16. P5 L16: Do you have any data on the occurrence or tendency for wildfires near this specific stations? If not, this comment should perhaps be removed.
**Response**: The sentence has been removed.

17. P9 Section 3.3 heading: I am a bit skeptical to the use of the word "Trend" in this heading and in the section text, as a trend can hardly be quantified based on a comparison between the years 2000 and 2010 (data for years 2008 and 2009 helps, but the data are scarce). Consider changing "trend" to "change" or something similar.
**Response**: It has been changed in Section3.3.

18. P9 L39: Please add a reference after "the past decade."
**Response**: The reference "Li et al., 2018" is added because it compares emission changes in China.

19. P11 L25: How would the results look if you use CAM-chem only for all the years?
**Response**: The numbers (in Table 2 for RBU scenario in 2008 and 2009 ) are indeed calculated with CAM-chem only.

20. P12 L16: ECE –> CEC?
**Response**: Yes, it has been changed to "CEC"

21. P13 L10: "The participating models … to 5.5%" can be removed as it has just been said above.
**Response**: It has been removed.

22. P13 L14: It says Frequency_Full_Impact15 twice :)
**Response**: The second one is removed.

23. P13 L34: Please give this in % change as well.
**Response**: The percent change is added.

24. P14 L37: Please add references after "recent years".
**Response**: The following references are added:
Chen, Y., Schleicher, N., Fricker, M., Cen, K., Liu, X. L., Kaminski, U., Yu, Y., Wu, X. F., and Norra, S.: Long-term variation of black carbon and PM2.5 in Beijing, China with respect to meteorological conditions and governmental measures, Environ Pollut, 212, 269-278, 10.1016/j.envpol.2016.01.008, 2016.
Feng, J. L., Zhong, M., Xu, B. H., Du, Y., Wu, M. H., Wang, H. L., and Chen, C. H.: Concentrations, seasonal and diurnal variations of black carbon in PM2.5 in Shanghai, China, Atmos Res, 147, 1-9, 10.1016/j.atmosres.2014.04.018, 2014.
Lu, Z., Streets, D. G., Zhang, Q., Wang, S., Carmichael, G. R., Cheng, Y. F., Wei, C., Chin, M., Diehl, T., and Tan, Q.: Sulfur dioxide emissions in China and sulfur trends in East Asia since 2000, Atmos Chem Phys, 10, 6311-6331, 10.5194/acp-10-6311-2010, 2010.
Zhu, J. L., Liao, H., and Li, J. P.: Increases in aerosol concentrations over eastern China due to the decadal-scale weakening of the East Asian summer monsoon, Geophys Res Lett, 39, L0980910.1029/2012gl051428, 2012.